# DNA methylation signature of chronic low-grade inflammation and its role in cardio-respiratory diseases

We performed a multi-ethnic Epigenome Wide Association study on 22,774 individuals to describe the DNA methylation signature of chronic low-grade inflammation as measured by C-Reactive protein (CRP). We find 1,511 independent differentially methylated loci associated with CRP. These CpG sites show correlation structures across chromosomes, and are primarily situated in euchromatin, depleted in CpG islands. These genomic loci are predominantly situated in transcription factor binding sites and genomic enhancer regions. Mendelian randomization analysis suggests altered CpG methylation is a consequence of increased blood CRP levels. Mediation analysis reveals obesity and smoking as important underlying driving factors for changed CpG methylation. Finally, we find that an activated CpG signature significantly increases the risk for cardiometabolic diseases and COPD.

In population-based studies, C-reactive protein (CRP)—an acute phase reactant—is commonly used as a proxy for chronic low-grade inflammation[1,2]. Low-grade inflammation plays a key role in the development of a wide range of disease such as type 2 diabetes (T2D)[3,4], coronary heart and other cardiovascular diseases[5], chronic obstructive pulmonary disease (COPD)[6] as well as several psychological disorders such as posttraumatic stress syndrome, schizophrenia and depression[7,8]. Prior studies have strongly linked elevated serum CRP to genetics[9], overweight and obesity[10,11] physical inactivity, fiber intake[12] and smoking[13,14]. However, the molecular mechanisms underlying the robust associations of CRP with these factors are not well-understood.

Epigenetic modifications such as the addition of a methyl group to a cytosine base of human DNA is commonly referred to as DNA methylation. This is a reversible process that affects gene expression. DNA methylation in white blood cells can be a consequence of exposure to risk factors such as subtle changes in regulatory regions of the genome in the setting of excess adiposity or smoking[15,16] or a consequence of diseases like global changes of CpG methylation in cancer[17]. These CpG methylation changes affect gene expression patterns of different human tissues[18], can be inherited across generations[19] and differ between ethnicites[18]. Studying differential DNA methylation in relation to chronic inflammation could highlight pathways that link the risk factors to diseases and adverse effects.

In this study we performed a multi-ethnic epigenome wide association study (EWAS) on serum CRP in 22,774 participants. The large sample size of our study allowed us to create a reference list of CpGs robustly associated to CRP. In addition, our strategy could be used as a blueprint for generation of robust marker sets for any exposure similar to the field of genome wide association analysis. Our study suggests DNA methylation as consequences of CRP, identifies underlying factors of the signature such as BMI and smoking and gives a measure of the contribution to development of disease such as T2D and coronary artery disease.

## Results

**Multi-ethnic discovery.** We found a total of 1765 markers significantly associated to serum CRP levels at a Bonferroni threshold ($P < 1e-7$) in our multi-ethnic meta-analysis (Fig. 1A) on 22,774 samples. Those were subdivided into African American and African Ancestries (AA), European Ancestries (EA) and South Asian Ancestries (SAA). Thirty studies provided summary statistics about the association of CpG methylation with serum CRP. Our regression model was adjusted for age, sex, blood cell type composition and technical confounders. Effect sizes (Supplemental Fig. 2) were ranging from $0.50 \pm 0.06$ to $9.85 \pm 0.29$ logarithmic mg/L change in CRP per unit increase in DNA methylation in blood (scale for methylation 0-1, where 1 represents 100% methylation). We also standardized the regression coefficients from the multi-ethnic discovery, for example, one CpG near NF-κB inhibitor epsilon (NFKBIE) showed a decrease of 8 standard deviations of DNA methylation per standard deviations of logarithmic CRP. Standardized regression coefficients were ranging from 22.2 for a CpG near STK40 to −21.5 for a CpG not in vicinity of any gene (Supplementary Data 2). $P$ values were ranging from $9.9 \times E-08$ to $1.9 \times E-69$.

We performed basic quality control of individual study DNA methylation association data, including replication within known CRP associated markers published by Ligthart et al.[20] as well as correlation of effect sizes between studies (Supplemental Figs. 3 and 4). Mean age of participants ranged from 16 (NFBC1986) to 75.5 (CHS-W) years, BMI from 23.7 kg/m² (NFBC1986) to 32.9 kg/m² (BHS-W Median serum CRP values extended from 0.2 mg/L mg/L (NFBC1986) to 4 mg/L mg/L (EPIC Norfolk). Across all studies, 49.3% of the participants were female (Table 1).

To prevent reporting false positive signals due to bias and unwanted variation such as population stratification, we applied Genomic Control (see "Methods" and Fig. 1B). Additionally, we applied an alternative strategy to control for bias and inflation[21] (Supplemental Figs. 8, 9 and Supplementary Data 2). Using this approach $P$-values were ranging from $2.2E-124$ to $1.9 \times E-06$, and we discovered a total of 144 additional CRP-associated markers given in Supplementary Data 3.

Next, we evaluated if the 1765 CRP-associated methylation markers were significant across three ancestries in our study (AA, EA, SAA), and whether there were ancestry-specific DNA methylation markers. We defined CpGs as replicating in ancestry-specific analysis if its $P$-value was below 0.05 and direction of effect was consistent in each ancestry-specific analysis. Among markers identified in the multi-ethnic meta-analysis, 1765) were significant and showed a consistent direction of effect in EA (100%, N~ 14,568, 22 studies,) alone, 1408 (79.7%, $N \sim 3430$, 3 studies) in AA and 1550 (87.8%, N ~ 2688,1 study) in SAA meta-analyses (Fig. 1C, Supplementary Data 2). In addition, to the 1765 markers discovered in multi-ethnic meta-analysis, we identified 62 Bonferroni significant markers in the EA only (Supplementary Data 4) and 2 markers significant in SSA ancestries only (Supplementary Data 4).

To further investigate differences between different ancestries, we plotted Z-scores across the ancestry-specific meta-analysis (Supplementary Fig. 4). However, we did not find any statistically significant evidence for heterogeneity in multi-ethnic meta-analysis.

**DNA methylation correlation structure.** To determine which of the 1765 CRP-associated markers were independent and which were correlated, we assessed the DNA methylation correlation structure of these 1765 markers across 4 studies ($N \sim 3920$). We noticed that the correlation structure across the analyzed cohorts were similar despite of differences in mean age, ranging from 16 to 61 years, and detection platform (Supplemental Fig. 8). Furthermore, we found that the correlation structure between CRP-associated CpGs was consistent across chromosome borders (Fig. 2A). However, the biggest contribution to the observed correlation structure was the physical proximity between 2 or more CpGs. To better understand this contribution to the correlation structure we binned the correlation values according to their distance separately for each chromosome and displayed the results as boxplot (Fig. 2B). The Pearson correlation coefficients depended on the distance between CpGs, with a drop below 0.2 at about 5 kb. This average correlation value stayed stable for quite a distance throughout the genome. Thus, we decided to apply a 5 kb window to identify a set of independent uncorrelated loci. This strategy restricted our list of 1765 marker to a list of 1511 loci, which henceforth was used as input for any further analysis.

Because we observed coherent DNA methylation patterns across chromosomes (Fig. 2A), we attempted to identify clusters within our set of 1511 independent loci. We used a density-based algorithm that suggested two correlation clusters within the analyzed data (Fig. 2C). Mapping these group assignments back to the original data (color code in Fig. 2A), we found that this assignment reflected the actual correlation values for most of the CpGs. Next, we looked into overlaps of the CpGs within each correlation cluster (Fig. 2C) with genomic features. We found similar distributions of the two clusters in broad genomic features such as CpG islands, gene bodies, distance to transcription start sites, and HiC. However, we found differences in more specific genomic features, we detected in 22% of cluster 1 CpGs being

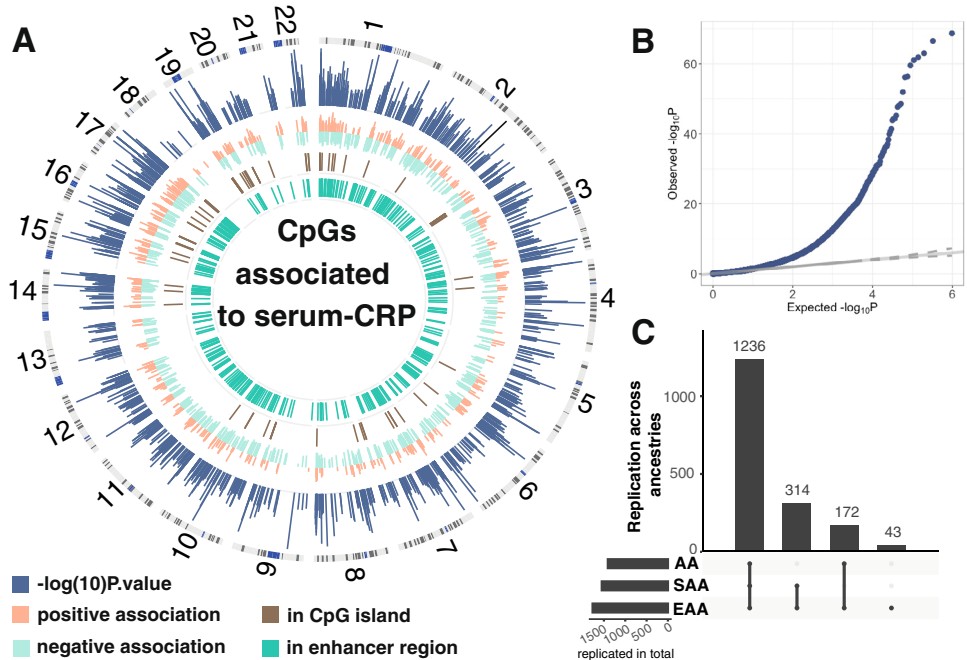

**Fig. 1 Results of multi-ethnic meta-analysis.** Panel **A** is a circos plot representation of the multi-ethnic meta-analysis. Outermost track is chromosome number followed by ideogram. The second track in blue is the Manhattan plot of the CpG CRP association results. Next track (in orange and light green) are effect sizes of CpG CRP associations, where orange represents positive associations and light green negative. Track represented in brown track color gives the overlap between the 1765 CRP associated CpG markers with CpG island in the genome. The innermost track (in dark green) gives the overlap with enhancer regions as defined by Roadmap project[23]. Panel **B** is a qqplot of the genomic control corrected *P*-values from the multi-ethnic meta-analysis. Panel **C** shows replication rates of 1765 across ancestries. Each bar gives the number of replicated CpGs across ancestries indicated as dots below the barplot. Horizontal bars reflect the total number of replicated CpGs per ancestry group.

*RELA* (also known as nuclear factor NF-kappa-B p65 subunit) binding sites as opposed to 7.8% in cluster2 CpGs (see below). Overlaps to GO terms suggest different functional contributions of the 2 correlation sets (see below). The DNA methylation risk score analysis, however, did not show significant different effects of cluster1 CpGs or cluster2 CpGs on clinically relevant phenotypes (Supplementary Data 23).

**Sensitivity analysis.** To evaluate the stability of the presented association results, we compared the base model to a model additionally adjusted for BMI across all studies. All 1765 markers showed consistent effect directions. The Pearson's correlation between Z-scores (denoted $\rho$ (Z-scores)) from these two models was 0.985 (Fig. 3A). Out of 187 published BMI-associated CpGs[16] we found 120 CpGs within our list of 1511 loci. In a subset of cohorts (see methods), we evaluated possible influences of other risk factors on the CpG CRP associations. For BMI adjusted model effect sizes were ranging from $0.241 \pm 0.151$ to $6.648 \pm 0.482$ logarithmic mg/L change in CRP per unit increase in DNA methylation in blood with consistent direction of effect. We did not observe any significant changes in the distribution of Z-scores when adjusting for additional risk factors such as smoking, lipids, insulin, BMI, hip circumference, and waist circumference. (see methods, Supplemental Fig. 9). We observed a total of 22 CpGs that changed the effect direction in at least one model in the sensitivity analysis. (Supplementary Data 5). A total of 140 CpGs had a nominal significant P value of heterogeneity, of which one marker had a Bonferroni significant *P*-value of heterogeneity.

**Driving forces of the CRP CpG association: Mendelian Randomization analyses.** CpG methylation can be a transient state.

Thus, we studied whether or not CpG methylation of the 1511 loci were causal for the altered serum CRP levels or if differences in CpG methylation is a consequence of altered CRP levels. It has been shown for BMI and Crohn's disease[16,22] that differences in CpG methylation were a consequence of the investigated trait. Applying a similar strategy, we performed a 2-sample Mendelian Randomization followed by a triangulation analysis ("Methods"). We combined genetic and epigenetic data from more than 7000 participants derived from 11 studies. We identified 709 valid genetic instruments for CRP associated CpG sites (Supplementary Data 6), and found 8 loci showing Bonferroni significant effects of their genetic instruments on CRP levels. Thus, suggesting a causal effect of these 8 CpGs on serum CRP levels. We further investigated the overall association between CpG instruments and serum CRP levels in a triangulation analysis, with the following basic assumption: If the effect of a CpG on serum CRP is causal, it is possible to predict the effect of the genetic instrument (CpG-specific SNP) on CRP via the combination of the effect of the same SNP on CpG methylation and CpG methylation on CRP. This association is shown as scatter plot. (Fig. 3C; Supplementary Data 7). The analysis did not suggest an overall causal effect of CpG methylation on serum CRP levels (Fig. 3C).

We further investigated if these 709 CpGs might be a consequence of altered serum CRP levels, but did not find Bonferroni significant associations between our genetic instruments for CRP and CpG methylation. This suggests no causal effects of CRP on any individual CpG (Supplementary Data 8). Triangulation analysis, however, revealed that the majority of CpGs predicted the observed effects. We observed a Pearson Rho of 0.17 ($P = 8.23e{-}06$) (Fig. 3D; Supplementary Data 8) and the sign test revealed a *P* value of 1.27e−05, suggesting a causal effect of serum CRP levels on the majority of CpGs.

**Table 1 Cohort characteristics.**

| Cohorts | N | Ethnicity | Age | Sex | CRP | BMI | Smoking |
|---|---|---|---|---|---|---|---|
| AIRWAVE | 1108 | EA | 41.6 (9.3) | 40 | 1.0 (2.8) | 27.2 (4.3) | 64.4 / 23.8 / 10.5 |
| ARIC | 2182 | AA | 56.1 (5.75) | 63.5 | 3.3 (7.7) | 30.1 (6.2) | 44.9 / 30.4 / 24.6 |
| ARIES | 777 | EA | 48 (4.28) | 100 | 1.0 (3.1) | 26.4 (5.1) | 43.0/27.2/5.4 |
| BHS-B | 246 | AA | 43.6 (4.5) | 45.1 | 2.1 (2.5) | 30.1 (6.9) | 54.2/ 21.1 / 24.7 |
| BHS-W | 572 | EA | 43.2 (4.5) | 39.8 | 3.0 (3.1) | 32.9 (8.9) | 49.8/ 16.1 / 34.1 |
| BIOS-CODAM | 160 | EA | 66.3 (6.8) | 46.2 | 2.2 (5.4) | 28.2 (4.2) | 26.3 / 58.1 / 15.6 |
| BIOS-LLS | 713 | EA | 58.9 (6.7) | 52.2 | 1.2 (5.5) | 25.1 (3.5) | 30.9 / 55.7 / 13.3 |
| BIOS-NTR | 894 | EA | 33.6 (15.1) | 65.9 | 1.4 (4.8) | 24.0 (4.0) | 57.2 / 24.8 / 17.9 |
| BIOS-PAN | 166 | EA | 63.2 (9.4) | 37.9 | 1.5 (6.3) | 25.6 (3.6) | 39.8 / 32.5 / 27.6 |
| CARDIOGENICS | 200 | EA | 56 (6.7) | 16.6 | 0.5 (6.4) | 27.7 (4.3) | 0.1759 / 82.41 / 0 |
| CHS-B | 321 | AA | 73.1 (5.5) | 62.3 | NA | 28.7 (5.2) | 44.8 / 37.4 / 54 |
| CHS-W | 321 | EA | 75.5 (5.1) | 60.4 | NA | 26.7 (5.0) | 44.2 / 41.4 / 11.8 |
| EstBB-CTG | 306 | EA | 50 (16.9) | 50 | 1.2 (4.4) | 26.4 (5.6) | 51.3 / 30.4 / 18.3 |
| EPIC Norfolk | 1278 | EA | 60 (8.8) | 50.9 | 4 (7.5) | 27.2 (4.4) | 45.1 / 39.3 / 15.6 |
| EPICOR | 507 | EA | 53.6 (7.3) | 37.9 | 1.1 (2.5) | 26.1 (3.9) | 36.5 / 30.9 / 32.5 |
| ESTHER-1a | 974 | EA | 62 (6.5) | 50.08 | 1.6 (5.6) | 27.1 (4.4) | 47.6/33.7/18.7 |
| ESTHER-1b | 543 | EA | 62 (6.6) | 61.5 | 2.2 (6.7) | 27.5 (4.8) | 47.3/34.6/18.1 |
| FHS | 2008 | EA | 66 (9) | 55 | 2.5 (2.9) | 28.2 (5.2) | / NA / 7.1 |
| GENOA-27k | 681 | AA | 65.1 (8.4) | 72.1 | 0.35(1.4) | 30.2 (6.5) | 58.8 / 27.6 / 13.5 |
| KORA | 1724 | EA | 61 (8.8) | 51 | 1.3 (3.7) | 27.5 (4.8) | 41.7/43.7/14.5 |
| LBC | 258 | EA | 72.1 (0.5) | 46.9 | 1.4 (3.5) | 27.6 (4.3) | 132/109/17 |
| LLD | 695 | EA | 45.3 (NA) | 58.2 | 1.7 (3.3) | 24.6 (4.2) | NA |
| LOLIPOP | 2688 | SAA | 50.3 (10) | 31.5 | 2.3 (7.2) | 27.1 (4.3) | 82.6/8.6/8.8 |
| NAS | 648 | EA | 73.2 (6.8) | 0 | 3.3 (6.1) | 28.1(4.1) | 29.1 / 66.7 / 4.1 |
| NFBC1966 | 727 | EA | 31 (0.33) | 56.1 | 0.7 (3.6) | 24.5 (3.5) | 51.7 / 21.3 / 25 |
| NFBC1986 | 517 | EA | 16 | 53 | 0.2 (3.4) | 23.7 (3.8) | 71.9 / 9.1 / 13.5 |
| ROTTERDAM | 722 | EA | 59.9 (8.2) | 53.7 | 2.6 (4.7) | 27.5 (4.8) | 28.8 / 44.1 / 27.1 |
| SHIP | 236 | EA | 51.5 (13.5) | 51.3 | 2.3 (4.0) | 27.1 (4) | 22.0 / 38.1 / 39.8 |
| TWINSUK | 416 | EA | 59.3(8.7) | 100 | 1.6(7.8) | 25.6(4.6) | 59.4/30.8/9.9 |
| YFS | 186 | EA | 44.2 (3.4) | 61 | 1.4(2.4) | 26.2 | NA |

Column Cohorts gives all cohorts participating in the multi-ethnic meta-analysis in alphabetical order. N is the number of informative samples for this analysis. EA combines all European ancestries, AA combines all African Ancestries and SAA combines all South Asian ancestries. Age is given in years plus standard deviation. Sex is given as percent female in every cohort. CRP is the median of measured serum CRP levels in each cohort. BMI is body mass index. Smoking status given in percent as never smokers / former smokers / current smokers.

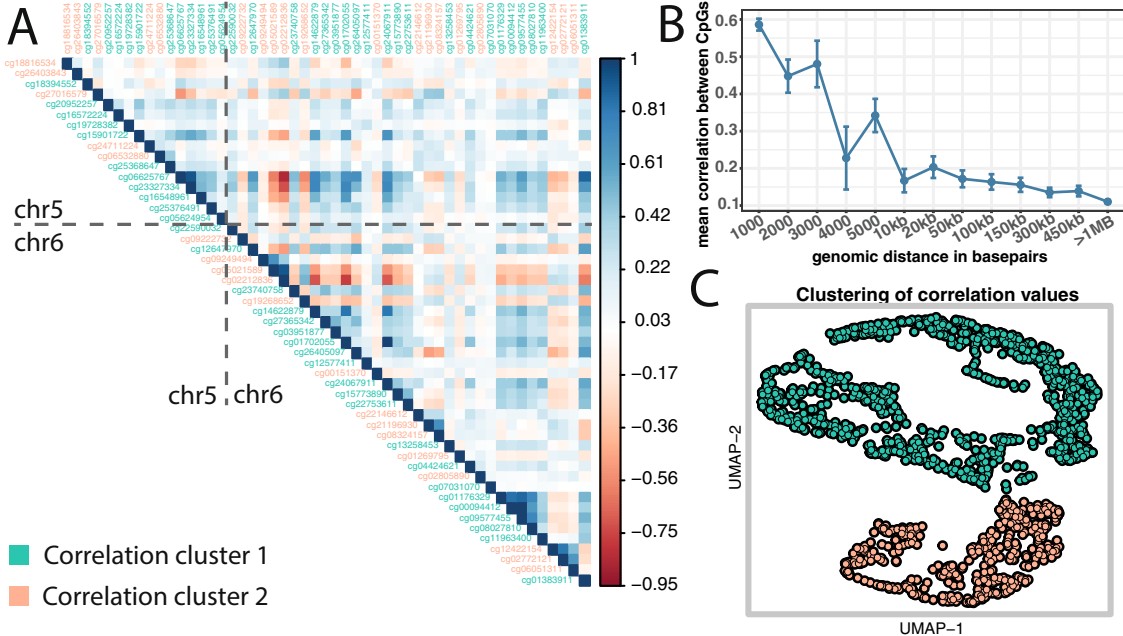

**Fig. 2 Correlation structure of CRP-associated CpG methylation.** Panel **A** gives meta-analyzed correlation values (Pearson Rho) across 4 cohorts. Displayed are all CRP-associated CpGs from a genomic region from chromosome 5 alongside with CRP-associated CpGs on chromosome 6. CpG ids are color-coded according to correlation clusters. Panel **B** CpGs correlation depending on distance. Correlation values were binned according to their distance to each other. X-axis gives distance between CpGs. Y-axis gives Pearson Rho observed each distance bin. In blue font, we plotted mean and standard errors of Person Rho values. Panel **C** is a UMAP representation of correlation values of the 1511 independent loci. Dots are color-coded according to their correlation cluster membership.

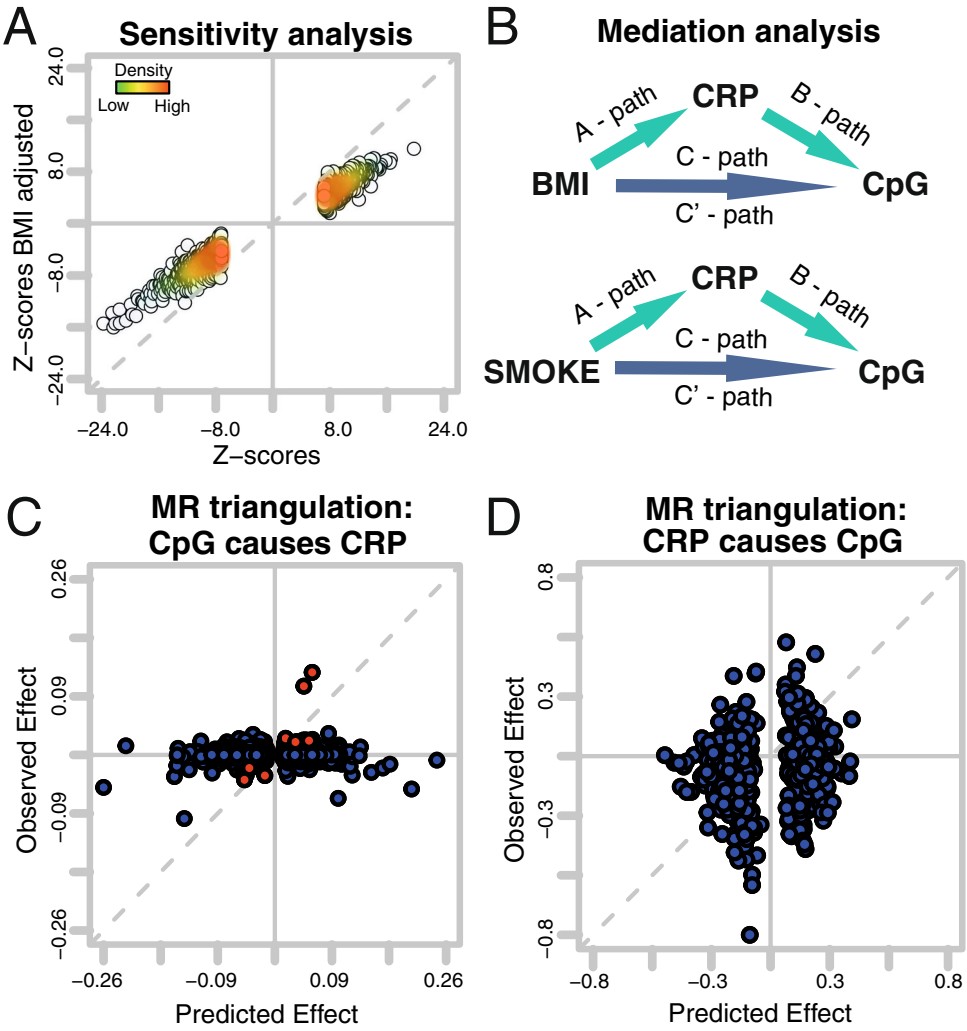

**Fig. 3 Driving forces of CpG signature.** Panel **A** is a comparison of Z-scores from the sensitivity analysis. Each dot represents a coefficient from the 1511 CRP-associated loci. The *X*-axis gives Z-scores derived from base model as applied in multi-ethnic meta-analysis. The *Y*-axis gives results from the same analysis adjusted for BMI. Panel **B** gives an overview of applied mediation analysis models. Panels **C** and **D** are representations of the Mendelian Randomization triangulation analysis. Each dot represents a CpG. The *Y*-axis is the observed effect, which is the association between the genetic instrument and outcome. The observed effects for Panel C originate from CpG instruments (SNPs) vs serum CRP levels. The predicted effect is the combined effect from the SNP CpG association and the CpG serum CRP association. Observed effects for **D** are the associations between a polygenic risk score for CRP (instruments) and CpG methylation. The predicted effects for panel D are the combined effects from the polygenic risk score for CRP (instruments) serum CRP association and serum CRP CpG methylation association (CRP$_{\text{Genetic risk score for CRP association}}$ × CRP $_{\text{CpG association}}$). The observed effect is the association of the polygenic risk score for CRP (instruments) to CpG methylation (CRP$_{\text{Genetic risk score for CpG association}}$).

**Driving forces of CRP CpG association: Mediation analysis.** Mendelian Randomization highlighted complex associations between serum CRP levels and CpG methylation. Sensitivity showed that BMI attenuates but not abrogates the effect of CRP on DNA methylation (Fig. 3) and a high overlap to published BMI-associated CpGs. We observed a similar situation for smoking (Supplementary Data 5). To better understand those relationships between these traits and our data, we performed a mediation analysis. Mediation analysis was first performed on a small subset of data testing 6 different models ("Methods", Supplementary Data 9), which pointed towards 2 models for further exploration (Fig. 3B): CRP being the mediator of a CpG methylation caused by BMI or by smoking (Fig. 3B). We used data from 4 cohorts (*N* ~ 3192) and performed mediation analyses according to Baron Kenney ("Methods", Fig. 3B). To avoid violation of assumption made by Baron Kenny approach we restricted the analysis to loci with a nominal significant C-path and A-path (Fig. 3) and

only report mediation if the indirect effect (C'-path–C-path) is negative. We found 1136 CpGs associated to BMI (nominally significant), of which 729 (64.1%) show a significant *P*-value for CRP mediating the effect of BMI effects on DNA methylation, with 213 loci reaching Bonferroni significance (Supplementary Data 2, 10).

We then evaluated possible effects of smoking on DNA methylation mediated by CRP (Fig. 3B). We found 386 markers associated to CRP also associated to smoking. Out of this set, we observed the effect of smoking on DNA methylation being mediated by CRP for 82 loci (21.2%) of which 32 loci reach Bonferroni significance levels (Supplementary Data 11).

**CRP associated CpGs in genomic and biological context.** Next, we evaluated if the CpG methylation signature was enriched within certain genomic features. We performed an over-representation analysis that mimicked the DNA methylation variation structure of our 1511 loci ("Methods"). We compared

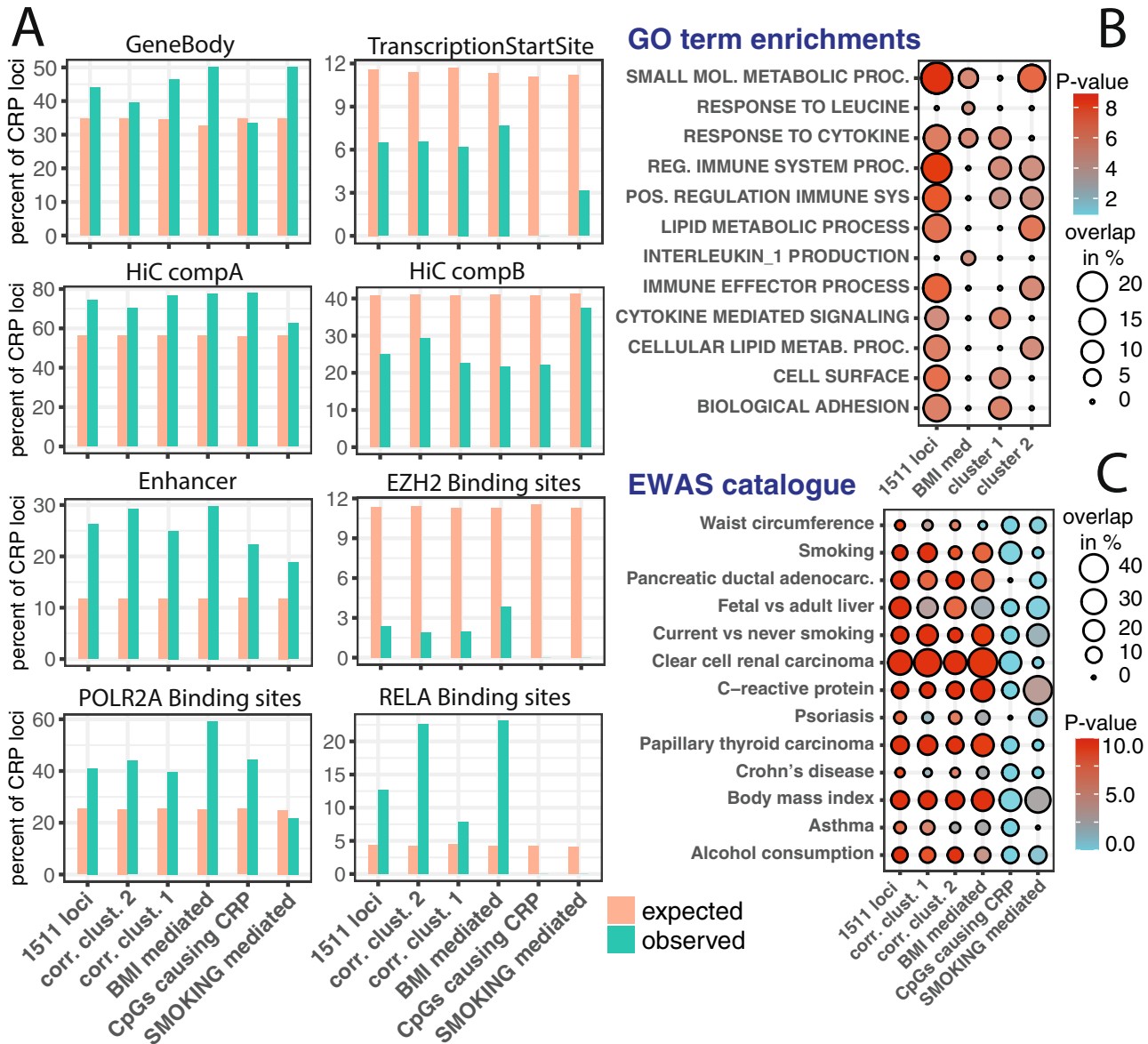

**Fig. 4 Overrepresentation analysis.** Panel **A** gives the percentage of each CRP-associated gene set that overlaps with selected genomic feature. Orange bars represent overlapping features by chance; green bars give the percentage that actually overlap with the CRP-associated CpGs. Transcription start site and enhancer genomic region were used as defined by the Roadmap project. HiC regions were as reported in GSE63525, where component A was connected to highly transcribed genomic regions and component B to heterochromatin. Panel **B** shows enrichment analysis between CRP-associated CpG that were significantly associated with mRNA expression. Empirical *P*-values for the overlap derived from a permutation test (described in more detail in method section "Overrepresentation analysis") are given as negative log10. Percent overlap indicates the percentage of CpGs present in each GO term set. Panel **C** gives overlaps between CpGs observed in this study and published gene lists from large scale EWAS.

the genomic positions of the CpG methylation signatures to histone modifications, DNase hypersensitivity, and chromatin model from the Roadmap project[23] as well as Hi-C data[24] and Encode Transcription factor binding sites[25](Fig. 4A). We found that CRP-associated CpGs were enriched within gene bodies but were depleted around transcription start sites (Fig. 4A; Supplementary Data 12). We further found CpG markers enriched in euchromatin and depleted in heterochromatin (Hi-C compartments, Supplementary Data 12, 13, 14, 15, 16, 17, 18). Analysis of Roadmap's chromatin model showed 30% of all CRP-associated CpGs situated within enhancer regions, whereas very few CpGs were on *EZH2* binding sites, indicating a minor impact of CpG islands on the presented DNA methylation signatures (Figs. 1, 4A).

We mapped the CRP methylation signature to histone marks across various tissues, however, did not find distinct evidence for one tissue driving the CRP CpG signatures (Supplemental Figs. 10, 11; Supplementary Data 12, 13, 14, 15, 16, 17, 18).

Mapping the CRP-specific DNA methylation signature to transcription factor binding sites revealed about 40% of loci being situated on Polymerase II subunit A binding site (*POLR2A*). This suggests DNA methylation is a key regulator of Polymerase II transcribed genes. Furthermore, we investigated the association between gene expression and CRP-specific CpG signature ("Methods"). Out of 1,511 CRP-associated loci 9% of CpGs were significantly associated with gene expression. 22 CpGs showed a positive association, whereas for the majority of CpGs (84 CpGs) an inverse relationship prevailed. This set of 106 CpGs was

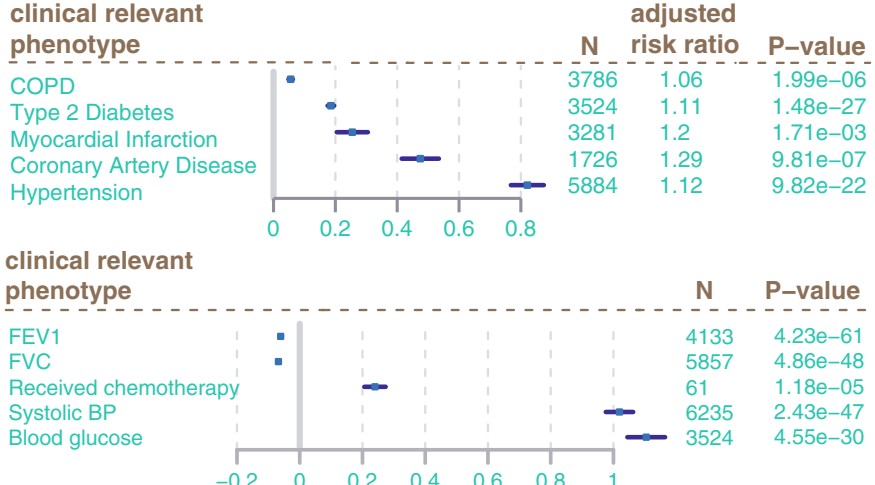

**Fig. 5 Associations of CRP DNA methylation signature to clinically relevant phenotypes.** Forest plots give estimate from logistic regression (logODDs) and confidence intervals (error bars) of CpG risk score regression against relevant phenotypes. N is the number of samples included in analysis. To produce adjusted relative risk estimates we transformed odds ratios as follows: RR = odds ratio/1 − (lifetime risk) + (life time risk × odds ratio). Those estimates indicate the theoretical maximum impact of the discovered CpG signature (100% DNA methylation change) on the tested traits. The risk conveyed by one percent change in the DNA methylation risk score on the tested traits was 1.007% for COPD, 1.7% for T2D, 2.9% for myocardial infarction 4.3% coronary artery disease, and 0.2% for hypertension. For continuous traits such as FEV1, FVC, systolic BP, and blood glucose estimates from linear regression including confidence intervals are given.

subjected to gene set enrichment analysis, which showed a mixture of enrichments for metabolic and immune system processes (Fig. 4B).

We also studied the overlap between CRP-associated CpGs and published gene lists in GWAS and EWAS catalog (Supplementary Fig. 9; Supplementary Data 20, 21, 22). Comparing the CRP signature to published EWAS results[26] we found significant overlaps to BMI, smoking, inflammatory diseases, and cancer-specific published gene lists (Fig. 4C; Supplementary Data 21 and 22).

**CRP associated CpGs in the clinical context.** Finally, we investigated the impact of CRP-associated CpGs on clinically relevant phenotypes. We created a beta weighted risk score, following the same approach as for the polygenic risk scores in GWAS analysis. Associations between CRP DNA methylation signatures and clinically relevant phenotypes were evaluated in the studies separately and then combined in a meta-analysis (Fig. 5). There were no associations between cancer and the CpG risk score, however, we found the CRP risk score positively associated to weight, BMI, and waist circumference (P < 6E−07; Supplementary Data 23). The CRP DNA methylation risk score was also positively associated to several other inflammation markers including IL6, IL1RA, and tumor necrosis factor receptor (TNFR) (P < 0.05, Supplementary Data 23). We found strong associations of CRP DNA methylation risk score to lung function (FEV1, FVC) as well as COPD and receipt of chemotherapy among breast cancer patients (Fig. 5B). Finally, we found strong associations of the CRP-associated methylation risk score with the tested cardiometabolic traits (Fig. 5A). We calculated the adjusted relative risk based on published life time risk for cardiometabolic traits ("Methods"). This may be interpreted as follows: A full activation of the CRP DNA methylation risk score as given in Fig. 5A indicates the theoretical maximum impact of the discovered CpG on these clinically relevant traits. For this, the CRP-associated CpGs would be either fully methylated or unmethylated according to their direction of effect. This is, however, very unlikely to happen for one single individual. Thus, we also calculated the risk conveyed by one percent change in the DNA

methylation risk score. The increased relative risk per one percent increased DNA methylation risk score was 1.007% for COPD, 1.7% for T2D, 2.9% for myocardial infarction, 4.3% coronary artery disease, and 0.2% for hypertension.

**Discussion**

With a sample size of 22,774 this epigenome wide association study on CRP, an important marker for chronic low-grade inflammation, is one of the largest EWAS efforts thus far. In this study, we identified a set of significantly associated CpGs that was 10× larger than in previously published EWAS[16,27,28]. Our analysis strategy in this study can be summarized as follows: To maximize power in this analysis we use a sparse regression model including only technical confounders. Next, we apply genomic control procedure, a conservative approach more typically performed in genome wide association studies[29] (Fig. 1B), to prevent from false positive associations between blood CRP levels and DNA methylation. This allows us to use the maximum available sample size in the regression analyses that can be taken forward in the downstream analyses. Within the study, we then work towards a better understanding of the driving forces of the CRP DNA methylation association. We perform Mediation and Mendelian Randomization analysis to understand the potential causal connection between DNA methylation, blood CRP levels, BMI, and smoking. Finally, we present a DNA methylation risk score analysis, without any underlying assumptions, based on all CRP-associated CpGs, alongside with more fine-grained, analysis stratified for the analysis results of the study (Supplementary Data 2). Applying this strategy, the presented marker sets replicated well across ancestries (Fig. 1C; Supplemental Fig. 7) and proved to be stable in various sensitivity analyses (Fig. 3A; Supplemental Fig. 9) and alternative meta-analysis strategies (Supplemental Fig. 4, Supplementary Data 3)

One of the most interesting questions in terms of DNA methylation is its causal vs. consequential role: does the observed changed DNA methylation pattern contribute to risk for the associated trait, or is it a consequence? Mendelian Randomization, in which we use single nucleotide polymorphisms (SNP) as proxies for the individual CpG or the investigated trait, can help

to answer this question[30]. If the SNPs associated with the CpGs discovered in this study are associated with CRP as well, we can infer a causal effect of the CpGs on chronic inflammation. In this case, DNA methylation would be the cause of changed serum CRP levels. Similar inference can be made with SNPs associated with chronic inflammation. This classical Mendelian Randomization can be extended to a triangulation analysis[31], which is especially useful for DNA methylation studies. In triangulation analysis, the effect size of the instrument-outcome association should match the combined effect size of the instrument-exposure association and exposure-outcome association. It has been shown that BMI as well as Crohn's disease cause DNA methylation changes[16,22]. Our Mendelian Randomization results suggest that the observed DNA methylation are likely a consequence of low-grade inflammation (Fig. 3D). These results, however, are less striking than similar results observed for BMI and Crohn's disease. This might be due to the increased number of markers evaluated in this study compared to previous studies or due to the complex relationship between low-grade inflammation and metabolic syndrome, smoking, stress, and many other factors[32].

To put DNA methylation into context with other risk factors for low-grade inflammation we performed a mediation analysis. We observed more Bonferroni significant markers than previous studies[33,34] by a factor of about 10, and assessed a total of 6 different mediation models ("Methods", Supplementary Data 9). We observed that a large proportion of the CRP-associated DNA methylation was caused by BMI (Fig. 3B) and a smaller fraction of the described CpG signature was caused by smoking.

The CRP DNA methylation signature includes a mixture of CpGs with evidence that they may play a central role in many cardiovascular and other diseases and CpGs that may be consequences of inflammation. One of the strongest markers for BMI CRP mediation is PHOSPHO1 (cg02650017). We found evidence that association of this marker, previously linked to BMI[16], is actually due to the effect of BMI on CRP (Supplementary Data 9, C2 path for cg02650017 not significant). Similarly, we discovered that MPRIP methylation (cg23842572), which was associated with smoking in a previous large-scale study associated to smoking[35], was also mediated by CRP. Interestingly, the very same CpG (cg23842572) is also associated with all-cause mortality[36].

Similar to other complex trait DNA methylation association studies we detected low-grade inflammation associated CpGs predominantly located in open chromatin structures[37,38], enhancers, and other regulatory regions in the genome and depleted in CpG island and related structures (Fig. 1A, 4). We replicated the published association between increased DNA methylation at AIM2 and lower serum CRP levels and lower expression of AIM2[20]. Additionally, our study suggests that this effect is due to a BMI → CRP → DNA methylation (Supplementary Data 9, C2 path for cg10636246 not significant). Furthermore, we observed decreased DNA methylation associated with higher levels of CRP and higher expression levels of NOD2, an established marker for chronic inflammation[39], which again was due to a BMI → CRP → DNA methylation mediation effect (Fig. 3B; Supplementary Data 9, cg01243823).

Our cluster analysis of correlation coefficients showed two distinct groups. We observed the most pronounced differences between correlation clusters within RELA binding site methylation (NF-κBp65 subunit, Fig. 4). GO term enrichment showed enrichment of Immune system processes for both clusters and enrichment of metabolic processes for cluster 2 (Fig. 4). A recent study suggests mutual regulation of overnutrition and inflammation by NF-κB[40], which fits very well to the GO term enrichments observed for cluster 2. Similarly, other studies showed that RELA binding sites are not only controlling Immune

system associated genes but can also control triglyceride levels and lipogenesis[41,42]. Finally, the vast majority of RELA binding sites is linked to epigenetic regulators[40] such H3K27ac, H3K4me both of which are closely linked to changes in DNA methylation[43,44]. Thus, we speculate that cluster 2 might reflect this mutual regulation of overnutrition and inflammation.

The connection between low-grade inflammation and adverse health outcomes is well established[45–47], thus we investigated if CpG methylation caused by low-grade inflammation explains these associations. On a global level, creating a DNA methylation risk score, we found strong evidence for this notion. However, individual DNA methylation changes also suggest this. In a large-scale nested case control study Chambers, JC et al.[48] identified 5 CpGs affecting T2D risk. All 5 CpGs were associated with CRP in this study and thus included in our risk score. The DNA methylation signal of two markers, PHOSPHO1 and SOCS3, were caused by a BMI → CRP → DNA methylation mediation. This highlights the important contribution of inflammation to T2D development and the importance of our findings to improve the understanding of existing data and how this dataset can serve as a reference for research into chronic inflammation. Similarly to T2D, we find four out of the six top markers associated with FEV1/FVC in a recent large scale meta-analysis[49]. The signal from one marker in the AHRR gene could be back traced to a smoking → CRP → DNA methylation mediation effect found in our study.

This study has limitations: We assayed DNA methylation in blood and thus we naturally investigated rather the consequences of changed CRP levels than the causes, as CRP is primarily synthesized in the liver[50] and adipose tissue[51]. Our study provides only a snapshot of DNA methylation changes associated with chronic inflammation, as we were limited to loci present on the Illumina Infinium Human Methylation450 Bead chips. The fact that more than 100,000 probes are situated on CpG islands[52], which as we and others[28,37] showed, further decreases the number of actual analyzed genomic loci. Due to limited data availability, we investigated only a small number of possible underlying reasons for the CRP-associated DNA methylation changes. Apart from smoking and BMI there might be other risk factors that actually drive the CRP signature[32]. This large scale setting, however, gives rise to other challenges, including the risk of false positives due to technical as well as unknown biological influences alongside with cellular heterogeneity[21], which can be amplified in larger samples.

The strengths of our study are its sample size and its multi-ethnic discovery combined with a stringent analysis controlling for unwanted variation, allowing conclusions relevant for public health and disease management to be drawn. The large number of samples guaranties stable, reproducible results. The novel discovered signature shows higher replication rates across ancestries than earlier EWAS on CRP with smaller sample sizes[20]. Furthermore, studies included in meta-analysis used different data normalization strategies and different microarray technologies, which makes the resulting DNA methylation signature very generalizable and likely to be reproducible in many contexts.

Since DNA methylation is a reversible process[53], our low-grade inflammation-associated DNA methylation signatures could serve as a valuable tool to monitor if changes in lifestyle are effectively decreasing the risk of adverse health outcomes. They could be used to monitor the efficacy of personalized interventions or may even pave the way for new epigenetic treatments. In a healthy population the DNA methylation signature can be applied as a proxy for chronic inflammation, and serve as an indicator for the transition between metabolically healthy obesity and obesity that most likely lead to adverse health outcomes.

In conclusion, we identified a robust set of CpGs associated with chronic inflammation in a large-scale multi-ethnic discovery analysis. Our analysis suggests that the discovered DNA methylation signature was largely a consequence of chronic inflammation that can be traced back to a mosaic of underlying driving factors such as smoking and BMI. The presented DNA methylation signature could be applied to understand the influence of low-grade inflammation in any existing or novel epigenome-wide association study and we hope that this large-scale meta-analysis inspires similar efforts to produce reliable signature for other complex traits. Most importantly, our study suggests that a sizable proportion of the impact from low-grade inflammation on the risk of heart disease, hypertension, T2D, and COPD is conveyed by DNA methylation, which implies that these effects might be reversible when changing the underlying causes such as BMI and smoking.

## Methods

**Study populations**. The study comprised of 22,774 participants from 30 independent studies. The 25 participating studies of European Ancestries are AIRWAVE, ARIES, BHS-W, BIOS-CODAM, BIOS-LLS, BIOS-NTR, BIOS-PAN, CARDIOGENICS, CHS-W, EstBB-CTG, EPIC Norflok, EPICOR, ESTHER-1a, ESTHER-1b, FHS, KORA, LBC, LLD, NFBC1966, NFBC1986, ROTTERDAM, SHIP-Trend, TWINS-UK, YFS; the four studies of African Ancestries (AA) are ARIC, BHS-B, CHS-B, GENOA; and the one study of South Asian Ancestries is LOLIPOP. Participants with log blood CRP levels outside median +/− 4× standard deviation were excluded from analysis.

Sensitivity analysis was performed in AIRWAVE, NFBC1966, NFBC1986 and LOLIPOP. Mediation analysis and correlation analysis were performed in AIRWAVE, NFBC1986, NFBC1966, and KORA. SNP associations for Mendelian Randomization analysis were retrieved from meta-analysis performed in AIRWAVE, KORA, NFBC1966, NFBC1986, and the BBMRI cohorts consisting of BIOS-CODAM, BIOS-LLS, BIOS-NTR, BIOS-PAN, ROTTERDAM, and LLD.

Analysis assessing association between clinical phenotypes and a CRP DNA methylation risk score were run in AIRWAVE, KORA, FHS, SHIP, and YFS.

**Methylation measurements and quality control**. DNA methylation was measured in whole blood using the Illumina450K or EPIC platform. Each cohort conducted their own quality control and normalization of DNA methylation data, as detailed in the Supplementary Data 1. The methylation beta (β) values were defined as $\beta = M/(M + U)$. Cohorts excluded DNA methylation markers above study specific detection $P$ value threshold (predominantly $P < 10^{-16}$) alongside with samples failing to produce a significant amount of markers below the detection $P$ value threshold (predominant call rate filter was 95%). Each cohort added technical covariates such as principal components of control probes, chip, chip row/column, bisulphite conversion batch etc. to the regression model as necessary in their dataset.

**Cohort-specific CRP DNA methylation associations**. Regressions were performed for each CpG individually using lm function as implemented in base R. The main regression model was:

*log(CRP) ~ DNAmeth + age + sex + estimated blood cell count + technical covariates*

For the sensitivity model, we added BMI to model and present results from this regression in the sensitivity analysis section. Overview of individual cohort covariates as well as cohort design, ethnicity, and sample numbers are given in Table 1. Blood cell count was estimated according to Housman et al.[54] as implemented in R package minfi[55].

**Meta-analysis and Genomic control procedure**. Meta-analysis was performed to combine results from all cohorts. We performed initial quality control and sanity checks individually on every study. In this context, replication rates of blood CRP markers reported in Ligthart S. et al.[20] as well as differences in effect sizes between studies were assessed (Supplementary Figs. 1 and 2). Meta-analysis was performed to identify DNA methylation markers associated to blood CRP levels. The analysis was restricted to autosomal markers on the Infinium Human Methylation 450K BeadChip. Further, we excluded probes if they: (i) had a SNP in last 10 bp of the probe sequence (ii) were flagged as cross-reactive probes[56]. This left 405,019 CpG sites for analysis in our meta-analysis. Effect sizes and standard errors of the 30 studies were combined using inverse variance weighting method as implemented in METAL software. We applied genomic control in and genomic control out to control for population stratification and other unmeasured confounding factors in the analysis[57]. Genomic control out was achieved by correcting $P$ values for the inflation factor lambda. Genomic control procedure was also applied in all sensitivity analyses and the multi-ethnic replication analysis. For the multi-ethnic

replication analysis meta-analysis was performed separately for each ancestry. We defined CpGs as replicating in ancestry-specific analyses if the $P$-value was below 0.05. As sensitivity analysis, we performed the meta-analysis using the R-package BACON[21] with default settings. To calculate standardized regression coefficients (normalization to standard deviations of lnCRP and DNA methylation) we combined the standard deviations of CpGs across all cohorts. Standard deviation of lnCRP were combined across 6 cohorts ($N = 7403$, ARIES, ARIC, EPICOR, LOLIPOP, NFBC1966, NFBC1988). Standard deviation of CpGs across all cohorts. Standardized regression coefficients were then calculated as follows $Coeff_{standardized} = (Coeff_{CRP \sim CpG + [..]} * combined\ SD_{CpG})/combined\ SD_{lnCRP}$.

**Correlation within CRP-associated markers**. Pearson correlation coefficients between DNA methylation vales were calculated separately for each cohort (AIRWAVE, NFBC1986, NFBC1986, KORA). Analysis was restricted to CRP-associated markers from multi-ethnic discovery meta-analysis. Correlation coefficients were then combined using the metacor function from the R package meta, which weights correlation coefficients based on sample size. Overall correlation structures between cohorts were very similar (Supplementary Fig. 3). To determine whether or not correlation of DNA methylation values is depending on genomic distance we binned correlation data into 13 bins depending on their distance to each CpG. Bins were: 1–5 kb in 1 kb intervals then 10, 20, 50, 100, 150, 300, 450, and >450 kb. Correlation between each CpG to any other CRP-associated CpG for every chromosome separately was combined and then represented as boxplot (Fig. 2). For any further analysis inclusive cluster analysis, we restricted the set of CpG($n = 1765$) markers to a set of 1511 independent CpG loci applying a 5 kb window (Supplementary Data 2). For this, we took the CpG marker with the lowest $P$-value for CPR association within a 5 kb forward to in depth analysis.

**Correlation cluster**. Correlation cluster analysis is based on meta-analyzed Pearson Rho values for 1511 independent CpG loci. We reduced correlation matrix to 2 UMAP dimension and calculated shared nearest neighbors using DBscan package. SNNclust density estimation was performed in a neighborhood size of $k = 35$, minimum points in cluster were 35, and eps parameter was 7.

**Sensitivity analysis**. To test the robustness of our CRP-associated markers and to find out whether or not the CRP-associated blood DNA methylation pattern may be driven by other metabolic risk factors we compared the summary statistics of the 1765 markers discovered with our base model "**CRP ~ DNA$_{meth}$ + age + sex + technical covariates + WBC estimates**" to the base model further adjusted with additional risk factors. To determine if a marker signal may be influenced by an additional risk factor we looked at 4 measures: 1. Does the direction of effect change; 2. Do Z-scores between models correlate; 3. Do we observe a significant difference in the coefficients of association (heterogeneity $P$-value); 4. Where possible: Is the marker part of a published list associated with the risk factor. BMI as additional risk factor was evaluated in all participating studies. Further risk factors were evaluated in a subset of studies (AIRWAVE, NFBC1966, NFBC1986, and LOLIPOP) Coefficients of subsequent models are compared to coefficients derived from the base model of those four studies. The following models were analyzed in our Sensitivity analysis. model 2s = base model + smoking status + packyears; model 3s = base model + waist circumference; model 4s = base model + hip circumference; model 5s = base model + total cholesterol; model 6s = base model + triglyceride; model 7s = base model + insulin; model 8s = base model + BMI + smoking status + packyears + waist circumference + hip.

**Residualisation of DNA methylation values**. For Mediation analysis, CRP DNA methylation risk score analysis as well as when looking for CpG instruments in our Mendelian randomization analysis we wanted to remove all unwanted variation from the DNA methylation values. To achieve this, we regressed out covariates known to influence the DNA methylation data from the quantile normalized DNA methylation beta values. The regression model was as follows:

DNAmeth ~ age + sex + CD4T + NK + Bcell + Mono + Neu + Eos + batch + [..]

For data integrated with CPACOR pipeline we added the first 10 principal components of the control probe PCs.

**Mendelian Randomization analysis**. We performed a Mendelian Randomization analysis to better understand the reasons for the differential DNA methylation associated to CRP. For this, we defined a set of 1511 differential methylated loci associated to blood CRP levels. This analysis was performed based on in genetic data retrieved from 11 participating cohorts: AIRWAVE, BIOS-CODAM, BIOS-LL, BIOS-LLS, BIOS-NTR, BIOS-PAN, BIOS-RS, NFBC1966, NFBC1986, and KORA totaling a sample number of $N \sim 7005$.

We performed analysis to investigate two Hypothesis:

1. DNA methylation is causal for CRP changes
2. DNA methylation is a consequence of changed CRP levels.

To test hypothesis 1, we needed to find instruments for DNA methylation. For this, we regressed DNA methylation value of every CpG against all SNPs present in

"cis" of the concordant CpG. The cis-region of every sentinel CpG was defined, as it's chromosomal position +/− 500 kb.

Regressions were performed using rvtests software. We applied a dominant inheritance model a using linear mixed model. The model included a kinship matrix to adjust for cryptic relatedness and population stratification, as well as estimated blood cell counts, age, and sex. Regressions were run individually in each cohort. Results were combined using METAL software in the inverse variance weighted mode. A valid instrument must have a genome wide significance SNP and CpG association ($P < 5 \times E{-08}$). For the association to the CRP we extracted coefficients and standard errors from the most recent CRP GWAS[9]. Furthermore, we excluded SNPs with a direct effect on the CRP. That is if SNP~CRP + lnCRP gives a significant association between SNP and lnCRP if we add CRP as covariate to the model. Again, this association were determined separately in every cohort using rvtests software and results were combined using METAL. A direct effect was defined as significant if the $P$ value of association was below Bonferroni threshold for 1511 tests ($3.3e{-05}$). We used the ratio method to determine significance in the Mendelian Randomization analysis as implemented in the R package MendelianRandomization where

$$\text{MRbeta} = \text{BETA}_{\text{CpG~SNP}} / \text{BETA}_{\text{CRP~SNP}}.$$

To test hypothesis 2, we used the latest published GWAS on CRP (Ligthart, 2018) to define a set of instruments for CRP. In contrast to testing of hypothesis 1 where we rely on large scale GWAS summary statistics for the association to our outcome (CRP). As there is currently no large scale GWAS summary statistics for Illumina 450k CpG as an outcome available, we created a CRP polygenic risk score starting 52 SNPs. To generate a beta weighted risk score we used PLINK version 1.9. Next, we regressed the risk score against every sentinel CpG under an additive model. CpG ~CRP$_{\text{PRS}}$ + age + sex + blood cell estimates + genetic PC [1..10]. Again, association testing was performed in every cohort separately, and results combined using METAL software. Furthermore, we excluded SNPs with a direct effect on the outcome (CpG). For direct effect estimation we regressed every CRP-associated SNP against DNA methylation levels of the 1511 sentinel CpGs adding CRP as covariate to the model. Bonferroni threshold for 52 SNP × 1511 CpGs ($P < 6.4 \, e{-07}$). All SNP associated with any CpG were excluded from the polygenic risk score. We found valid instruments for 709 CpGs which were used in this Mendelian Randomization. To retrieve a comparable result for both Hypothesis, we also restricted the CRP causes CpG changes direction of the MR to 709 CpGs. We used the ratio method to determine significance in the Mendelian Randomization analysis as implemented in the R package MendelianRandomization where

$$\text{MRbeta} = \text{BETA}_{\text{CRP~SNP}} / \text{BETA}_{\text{CpG~SNP}}.$$

**Mendelian Randomization triangulation**. Next, we wanted to understand if there is general trend for all CpGs to be cause or consequence of blood CRP levels. For this, we applied a triangulation approach: The effect of the instrument on the outcome (observed effect) should equal the product of the effect of the instrument on the exposure and the effect of the exposure on the outcome. For hypotheses 1, the observed effect is CRP ~SNP and the predicted effect is the product of the effects of DNA methylation on the SNPs and CRP on DNA methylation. For hypotheses 2 the observed effect is DNAmethylation ~GRS (polygenic CRP risk score) and the predicted effect is the product of effect of CRP ~GRS and CRP ~DNAmethylation. If the observed effect is mediated through the predicted effect those will correlate.

**Mediation analysis**. We performed a mediation analysis to better understand our CRP-associated DNA methylation markers. We followed the analysis framework as suggested by Baron Kenny (Fig. 3B). We assessed the results of four regressions:

- path: CRP ~ BMI + covariates,
- b-path: DNAmeth ~ CRP + BMI + covariates,
- c-path: DNAmeth ~ BMI + covariates (total effect) and
- c'-path: DNAmeth ~ BMI + CRP + covariates.

From this we can assess the indirect effect which is $a*b$. Values of this multiplication were compared to $c - c'$, which should give the same result. Next, we performed an Aroian Sobel test to assess the significance of the indirect effect. Briefly, we extracted coefficients and standard errors from a path regressions and b-path regressions and calculated a $Z$ score as follows: $Z_{\text{score}} = (a*b)/\sqrt{((b^2*SE_a^2) + (a^2*SE_b^2)) + (SE_a*SE_b)}$.

This was performed for every CpG separately. Z score was calculated only if the indirect effect was negative and we saw a significant association ($P < 0.05$) between BMI and the tested CpG. For this analysis we excluded all samples with BMI values outside of mean (BMI) +/− 4*SD(BMI). We tested 6 different mediation models in NFBC1966. Model 1: BMI -> CRP -> DNAmeth; Model 2: CRP -> BMI -> DNAmeth; Model 3: SMOKING -> CRP -> DNAmeth; Model 4: BMI -> DNAmeth -> CRP; Model 5: CRP -> DNAmeth -> BMI; Model 6: SMOKING -> DNAmeth -> CRP (Supplemental results Table 1). For model selection, we evaluated if one or more assumption for Mediation analysis according to Baron Kenney were violated. As given in Supplementary Data 10 we did not find any significant association in path A for model 2; model 4 and 5 did not have a negative indirect effects and model 6 did not produce results, because path C association between Smoking and CRP was not significant in NFBC1966. That left only model

1 and model 3 that did not violate assumption necessary to perform a meaningful mediation analysis. Additionally, incorporating published literature into the model selection process we were looking into studies that showed that BMI is causal for increased levels of blood CRP[10,58], on a global level BMI causes alterations in DNA methylation patterns[16] as well as CRP causing changes in alteration in DNA methylation pattern (this study). This makes models 2, 4, 5, and 6 highly unlikely as at least one of their path according to Baron Kenny is in conflict with the findings of these studies.

Thus, we extended the analysis to AIRWAVE, NFBC1986 and KORA cohort ($N \sim 3192$) for model 1 and model 3. To achieve this we ran all regressions (pathA to pathC') separately for each cohort and combined the effect sizes and standard errors using inverse variance weighted approach. The meta-analyzed results were then the basis for Mediation analysis according to Baron Kenny.

**Association to Gene-expression**. In Framingham Heart study a total of $n = 2,648$ participants had DNA methylation data (Illumina 450k array) and Gene expression data (Affymetrix Human Exon Array) available. The quality control process is described in a published manuscript[59]. To retrieve association between expression and DNA methylation we performed linear regression analysis Geneex ~ CpG + sex + age + houseman celltype + Principal Components and other technical covariates, accounting for familial relationship.

Out of the 1511 CpG loci we found that 1320 could be mapped to EntrezID in cis. From this pool of CpG 1170 could be mapped to the transcript IDs used in the CpG × Gene expression data set. This gives a Bonferroni threshold of $4.3e{-05}$ that corrects for 1170 tests. Bonferroni significant association are given in Supplementary Data 19.

**Overrepresentation analysis**. We wanted to determine the overlaps between our 1511 CRP-associated loci and publicly available lists of genomic features such as Histone Modifications, Transcription factor binding sites, and chromatin states. For this, we performed permutation tests. First, we assessed the number of overlaps between our 1511 CRP-associated markers and a genomic feature of interest. Then we sampled 10,000 sets of 1511 markers. For every set of 1511 marker we recorded the number of overlaps to the genomic feature of interest. Those 10.000 overlaps created our H0 distribution. We calculated an empirical P values separately based on either the number of entries with higher or the number of entries with lower overlap in our H0 distribution compared to our observed number of overlaps recorded within the 1511 CRP-associated loci. Additionally, we calculated a Fisher $P$ value based on the mean of our H0 distribution and the observed overlap. We collected the mean and standard errors of for each CpG from most studies in this EWAS. 90% of standard errors were between 0.007 and 0.075. SD values in this range were binned the SD data into 0.005 intervals. For generation of 10000 random sets of markers we used the equal numbers of markers within each standard error bin as observed in the CRP-associated loci.

We calculated overlaps to the CpG annotation as given in the Illumina Manifest file. We retrieved DNaseI-accessible sites from Encode project (http://www.uwencode.org/proj/hotspot), specifically gapped peaks from Release 9 called with MACSv2.0.10. For histone marks (H3K9 and H3K27) were retrieved gapped peak data from Roadmap project for a collection of 127 tissues and cell lines as well as probabilities for Roadmap 15 state chromatin model for selected cell lines[23]. Encode Transcription factor binding sites were retrieved from UCSC browser http://genome.ucsc.edu/cgi-bin/hgTrackUi?db=hg19&g=wgEncodeRegTfbsClusteredV3. Next, we mapped Illuminas 450k probes to the GWAS catalog hits (downloaded 20170626) of traits having 50 or more hits recorded in the catalog. We allowed a window of 1MB to match the 450k probes to each GWAS catalog entry and removed one or more hits in case they were overlapping. Finally, we retrieved chromosome conformation capture data (HiC) as described by Bing Ren et al.[60] Supplementary Figures 10 and 11 give a representative result for two enrichment analyses. Over representation analysis was performed with several subsets of the data: all loci, BMI mediated loci, smoking mediated loci, loci that cause CRP changes, correlation cluster 1 and cluster2.

**Association to clinical phenotypes**. We calculated one risk score per subset of loci in AIRWAVE, ARIES, FHS, KORA, SHIP-Trend, and YFS. A total of 7 risk scores per phenotype was produced. Those are: all loci, BMI mediated loci, smoking mediated loci, CpGs identified to affect serum CRP levels, correlation cluster 1 loci, correlation cluster 2 loci, and loci that are also associated with changes in gene expression in cis. We calculated a beta weighted risk score using the coefficients from the multi-ethnic discovery analysis (similar to a polygenic risk score in GWAS).

For every participant in each study:

$$\text{CpG}_{\text{riskSCORE}} = \Sigma \, \text{CpG}_{\text{methylation}} \times \text{Coefficient}_{\text{transEthnicDiscovery}}$$

Then we performed logistic regression model (glm() option in R) for each CpG$_{\text{riskSCORE}}$ against each outcome. Depending on the availability of phenotypes across cohorts we combined the effect sizes and standard errors using inverse variance weighted approach as implemented in METAL software. We present meta analyzed effect sizes (logODDs) and P-values from logistic regression. Odd ratios were transformed to produce adjusted relative risk estimates:

$$RR = \text{odds ratio}/1 - (\text{lifetime risk}) + (\text{life time risk} \times \text{odds ratio}).$$

We used lifetime risk estimates from current literature: COPD 11.45%[61], T2D 39.9%[62], MI 24.8%[63], CAD 40.15%[64] and Hypertension 81%[65].

**Reporting summary**. Further information on research design is available in the Nature Research Reporting Summary linked to this article.

## Data availability

The individual participant data included in this project are generally not publicly available due to data protection laws, but can be applied from the individual studies on reasonable request. Information about the individual studies analyzed in this manuscript can be found in the supplementary information. The summary statistics from the meta-analyses are available on figshare website (https://doi.org/10.6084/m9.figshare.19188674.v1).

## Code availability

All code for data cleaning and analysis associated with the current submission is available at https://github.com/Mwielscher/EWAS_CRP.

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

## Acknowledgements

Acknowledgements and funding sources are listed in the Supplementary Information.

## Author contributions

First draft of manuscript and scripting all analysis codes: M.W., Writing team: M.W., A.D., M.-R.J., and P.R.M.; sensitivity anlaysis: M.W., R.M., and O.R.; Mendelian randomization and mediation analysis: M.W., P.R.M., B.K., R.M., I.Q.C.; and correlation analysis: M.W., B.B.; gene-experssion analysis: R.J.; risk score analysis: M.W., B.K., R.J., K.N.C., P.P.M., E.W., O.R., and A.T. Management and data analysis of individual studies and critical review of manuscript: M.W., P.R.M., B.K., R.J., R.M., O.R., Y.Z., B.B., E.W.[9,10], P.P.M., P.S., R.W., P.-C.T., S.P., R.E.M., G.F., G.C., V.K., M.G., B.M.P., M.L., J.C.B., B.L., N.S., I.J.D., M.C.-H., J.A.B., A.C., E.S., A.K.S., A.H.M., M.A.T., E.M., X.G., J.B.J.v.M., J.G.-S., W.R., W.K., A.P., W.W., M.F., T.Z., W.C., Y.X., A.T., M.N., H.J.G., M.D., T.L., W.G., L.M., T.T., K.F., L.L.W., S.K., P.V., N.V., P.v.d.H., L.I., C.S., S.P., V.K., R.T., E.T., G.M., M.A.H., O.T.R., E.C., A.A.B., M.K., K.-H.H., S.L., K.N.C., J.S.K., A.K., B.T.H., P.D., C.R., K.K.O., J.T.B., E.B., P.E., H.B., M.B., D.L., M.W., J.C.C., A.D., M.-R.J.

## Competing interests

H.J.G. has received travel grants and speakers honoraria from Fresenius Medical Care, Neuraxpharm, Servier, and Janssen Cilag as well as research funding from Fresenius Medical Care. All other study authors declare no competing interests.

## Additional information

Matthias Wielscher [1,2✉], Pooja R. Mandaviya[3,4], Brigitte Kuehnel[5], Roby Joehanes[6,7], Rima Mustafa [1], Oliver Robinson [1], Yan Zhang[8], Barbara Bodinier [1], Esther Walton [9,10], Pashupati P. Mishra[11,12,13], Pascal Schlosser [14], Rory Wilson [5], Pei-Chien Tsai[15,16], Saranya Palaniswamy [1,17], Riccardo E. Marioni[18], Giovanni Fiorito[19,20], Giovanni Cugliari[21], Ville Karhunen [1], Mohsen Ghanbari [22,23], Bruce M. Psaty [24,25,26], Marie Loh [27,1], Joshua C. Bis [28], Benjamin Lehne[1], Nona Sotoodehnia[28], Ian J. Deary[29], Marc Chadeau-Hyam [1], Jennifer A. Brody [28], Alexia Cardona [30], Elizabeth Selvin[31,32], Alicia K. Smith [33], Andrew H. Miller [34], Mylin A. Torres[35], Eirini Marouli [36], Xin Gào [8], Joyce B. J. van Meurs[3], Johanna Graf-Schindler[5], Wolfgang Rathmann[37], Wolfgang Koenig [38,39,40], Annette Peters [5,39], Wolfgang Weninger[2], Matthias Farlik [2], Tao Zhang[41], Wei Chen [42], Yujing Xia [15], Alexander Teumer [43,44], Matthias Nauck [44,45], Hans J. Grabe[46], Macus Doerr [44,47], Terho Lehtimäki [11,12,13], Weihua Guan[48], Lili Milani [49], Toshiko Tanaka[50], Krista Fisher [49,51], Lindsay L. Waite[52], Silva Kasela[49], Paolo Vineis[1,20], Niek Verweij[53], Pim van der Harst [54], Licia Iacoviello[55,56], Carlotta Sacerdote[57], Salvatore Panico[58], Vittorio Krogh [59],

Rosario Tumino[60], Evangelia Tzala[1], Giuseppe Matullo[61,62], Mikko A. Hurme[63], Olli T. Raitakari[64,65,66], Elena Colicino[67], Andrea A. Baccarelli[68], Mika Kähönen[12,69], Karl-Heinz Herzig [70,71], Shengxu Li [72], BIOS consortium, Karen N. Conneely[73], Jaspal S. Kooner[74,75], Anna Köttgen [14,31], Bastiaan T. Heijmans [76], Panos Deloukas [36], Caroline Relton[10], Ken K. Ong [30], Jordana T. Bell [15], Eric Boerwinkle[77,78], Paul Elliott [1,79,80,81], Hermann Brenner [8,82], Marian Beekman [83], Daniel Levy[6,7], Melanie Waldenberger [5,39], John C. Chambers [1,27,75], Abbas Dehghan [1,20,84] & Marjo-Riitta Järvelin [1,17,85,86]✉

[1]Department of Epidemiology and Biostatistics, School of Public Health, Imperial College London, London, UK. [2]Department of Dermatology, Medical University of Vienna, Vienna, Austria. [3]Department of Internal Medicine, Erasmus University Medical Center, Rotterdam, The Netherlands. [4]Maastricht Centre for Systems Biology (MaCSBio), Maastricht University, Maastricht, The Netherlands. [5]Research Unit Molecular Epidemiology, Institute of Epidemiology, Helmholtz Zentrum München, German Research Center for Environmental Health, Neuherberg, Bavaria, Germany. [6]Framingham Heart Study, Framingham, MA, USA. [7]Population Sciences Branch, National Heart, Lung, and Blood Institute, National Institutes of Health, Bethesda, MD, USA. [8]Division of Clinical Epidemiology and Aging Research, German Cancer Research Center, Heidelberg, Germany. [9]Department of Psychology, University of Bath, Bath, UK. [10]Medical Research Council Integrative Epidemiology Unit, Bristol Medical School, University of Bristol, Bristol, UK. [11]Department of Clinical Chemistry, Faculty of Medicine and Health Technology, Tampere University, Tampere, Finland. [12]Finnish Cardiovascular Research Centre, Faculty of Medicine and Health Technology, Tampere University, Tampere, Finland. [13]Department of Clinical Chemistry, Fimlab Laboratories, Tampere, Finland. [14]Institute of Genetic Epidemiology, Faculty of Medicine and Medical Center - University of Freiburg, Freiburg, Germany. [15]Department of Twin Research and Genetic Epidemiology, King's College London, London, UK. [16]Department of Biomedical Sciences, Chang Gung University, Taoyuan City, Taiwan. [17]Center for Life Course Health Research, Faculty of Medicine, University of Oulu, Pentti Kaiteran katu 1, Linnanmaa, Oulu, Finland. [18]Centre for Genomic and Experimental Medicine, University of Edinburgh, Edinburgh, UK. [19]Laboratory of Biostatistics, Department of Biomedical Sciences, University of Sassari, Sassari, Italy. [20]MRC Centre for Environment and Health, School of Public Health, Imperial College London, London, UK. [21]Fondazione IRCCS Istituto Nazionale dei Tumori, Milan, Italy. [22]Department of Epidemiology, Erasmus MC University Medical Center, Rotterdam, The Netherlands. [23]Department of Genetics, Mashhad University of Medical Sciences, Mashhad, Iran. [24]Cardiovacular Health Research Unit, Department of Medicine, University of Washington, Seattle, WA, USA. [25]Department of Health Services, University of Washington, Seattle, WA, USA. [26]Department of Epidemiology, University of Washington, Seattle, WA, USA. [27]Lee Kong Chian School of Medicine, Mandalay Road, Singapore, Singapore. [28]Cardiovascular Health Research Unit, Division of Cardiology, University of Washington, Seattle, WA, USA. [29]Lothian Birth Cohorts, Department of Psychology, University of Edinburgh, Edinburgh, UK. [30]MRC Epidemiology Unit, School of Clinical Medicine, University of Cambridge, Cambridge, UK. [31]Dept. of Epidemiology, Johns Hopkins Bloomberg School of Public Health, Baltimore, MD, USA. [32]Welch Center for Prevention, Epidemiology, and Clinical Research, Johns Hopkins Bloomberg School of Public Health, Baltimore, MD, USA. [33]Departments of Gynecology and Obstetrics & Psychiatry and Behavioral Science, Emory University School of Medicine, Atlanta, GA, USA. [34]Department of Psychiatry and Behavioral Sciences, Emory University School of Medicine, Atlanta, Georgia. [35]Department of Radiation Oncology, Winship Cancer Institute, Emory University School of Medicine, Atlanta, GA, USA. [36]William Harvey Research Institute, Barts and The London School of Medicine and Dentistry, Queen Mary University of London, London, UK. [37]Institute for Biometrics and Epidemiology, German Diabetes Center, Leibniz Center for Diabetes Resesarch at Heinrich Heine University Düsseldorf, Düsseldorf, Germany. [38]Deutsches Herzzentrum München, Technische Universität München, Munich, Germany. [39]German Center for Cardiovascular Research (DZHK), Partner Site Munich Heart Alliance, Munich, Germany. [40]Institute of Epidemiology and Medical Biometry, University of Ulm, Ulm, Germany. [41]Deptarment of Biostatistics, School of Public Health, Shandong University, Jinan, China. [42]Department of Epidemiology, School of Public Health and Tropical Medicine, Tulane University, New Orleans, LA, USA. [43]Institute for Community Medicine, University Medicine Greifswald, Greifswald, Germany. [44]DZHK (German Center for Cardiovascular Research), Partner Site Greifswald, Greifswald, Germany. [45]Institute of Clinical Chemistry and Laboratory Medicine, University Medicine Greifswald, Greifswald, Germany. [46]Department of Psychiatry and Psychotherapy, University Medicine Greifswald, Greifswald, Germany. [47]Department of Internal Medicine B, University Medicine Greifswald, Greifswald, Germany. [48]Division of Biostatistics, School of Public Health, University of Minnesota, Minneapolis, MN, USA. [49]Estonian Genome Centre, Institute of Genomics, University of Tartu, Tartu, Estonia. [50]Translational Gerontology Branch, Biomedical Research Center, National Institute on Aging, National Institutes of Health, Baltimore, MD, USA. [51]Institute of Mathematics and Statistics, University of Tartu, Tartu, Estonia. [52]HudsonAlpha Institute for Biotechnology, Huntsville, AL, USA. [53]Department of Cardiology, University of Groningen, University Medical Center Groningen, Hanzeplein 1, Groningen, The Netherlands. [54]Department of Cardiology, Division of Heart and Lungs, University Medical Center Utrecht, Utrecht, The Netherlands. [55]Department of Epidemiology and Prevention, IRCCS NEUROMED, Pozzilli, IS, Italy. [56]Department of Medicine and Surgery, Research Center in Epidemiology and Preventive Medicine (EPIMED), University of Insubria, Varese-Como, Italy. [57]Unit of Cancer Epidemiology, Citta' della Salute e della Scienza Hospital and Centre for Cancer Prevention, Turin, Italy. [58]Dipartimento di Medicina Clinica e Chirurgia Federico II University, Naples, Italy. [59]Epidemiology and Prevention Unit, Fondazione IRCCS Istituto Nazionale dei Tumori, Milan, Italy. [60]Cancer Registry and Histopathology Department, "Civic - MPP Arezzo" Hospital, ASP Ragusa, Ragusa, Italy. [61]Department of Medical Sciences, University of Turin, Turin, Italy. [62]AOU Città della Salute e della Scienza di Torino, Torino, Italy. [63]Faculty of Medicine and Health Technology, Tampere University, Tampere, Finland. [64]Research centre of Applied and Preventive Cardiovascular Medicine, University of Turku, Turku, Finland. [65]Department of Clinical Physiology and Nuclear Medicine, Turku University Hospital, Turku, Finland. [66]Centre for Population Health Research, University of Turku and Turku University Hospital, Turku, Finland. [67]Department of Environmental Medicine and Public Health, Icahn School of Medicine at Mount Sinai, New York, NY, USA. [68]Laboratory of Environmental Epigenetics, Departments of Environmental Health Sciences and Epidemiology, Columbia University Mailman School of Public Health, New York, NY, USA. [69]Department of Clinical Physiology, Tampere University Hospital, Tampere, Finland. [70]Research Unit of Biomedicine, Medical Research Center, Faculty of Medicine, University of Oulu, and Oulu University Hospital, Oulu, Finland. [71]Department of Gastroenterology and Metabolism, Institute of Pediatrics, Poznan University of Medical Sciences, Poznan, Poland. [72]Children's Minnesota Research Institute, Children's Minnesota, Minneapolis, MN, USA. [73]Department of Human Genetics, Emory University School of Medicine, Atlanta, GA, USA. [74]National Heart and Lung Institute, Imperial College London, London, UK. [75]Department of Cardiology, Ealing Hospital, London North West Healthcare NHS Trust, Southall, UK. [76]Molecular Epidemiology, Department of Biomedical Data Sciences, Leiden University Medical Center, Leiden, The Netherlands. [77]Human Genetics Center, University of Texas Health Science Center at Houston, Houston, TX, USA. [78]Human Genome Sequencing Center, Baylor College of Medicine,

Houston, Houston, TX, USA. [79]MRC Centre for Environment and Health, School of Public Health, Imperial College London, London, UK. [80]Imperial Biomedical Research Centre, Imperial College London, London, UK. [81]British Heart Foundation, BHF, Centre for Research Excellence, Imperial College London, London, UK. [82]Network Aging Research, Heidelberg University, Heidelberg, Germany. [83]Department of Medical Statistics and Bioinformatics, Leiden University Medical Center, Leiden, The Netherlands. [84]UK Dementia Research Institute at Imperial College London, London, UK. [85]Unit of Primary Care, Oulu University Hospital, Oulu, Finland. [86]Department of Life Sciences, College of Health and Life Sciences, Brunel University London, London, UK. A list of authors and their affiliations are provided in the Supplementary Material.
✉email: matthias.wielscher@meduniwien.ac.at; m.jarvelin@imperial.ac.uk

