## [Peer Review File · Nature Communications]

DNA methylation signature of chronic low-grade inflammation and its role in cardio-respiratory diseasesReviewers' Comments:

Reviewer #1:

Remarks to the Author:

The authors present an interesting analysis of a very large sample size, where they demonstrate relationships between CRP with DNA methylation, then perform additional analyses to assess potential mediation, confounding, MR, gene-expression effects, and associations with downstream factors (clinical phenotypes). They identify a list of ~1,500 CpGs that are strongly associated with CRP, and ... There are some powerful aspects of this study, including the very large sample size and very thorough analysis. However, there are some questionable analyses and weaknesses that dampen my enthusiasm.

Major:

1. The authors applied genomic control to deal with population stratification and other unmeasured confounding. However, while GC does decrease inflation by restricting p-value thresholds, it does not correct for bias such as confounding. Since confounding is not addressed by GC, the authors should apply an alternative approach for addressing residual confounding that was specifically developed for addressing inflation and bias in EWAS (Controlling bias and inflation in epigenome- and transcriptome-wide association studies using the empirical null distribution | Genome Biology). This should be applied to cohort-specific results prior to fixed-effects meta-analysis.
2. However, the best approach is to control for confounding within the regression model itself, as long as the appropriate data are available, which raises another question. Why did the authors not adjust for BMI and smoking (or other variables in the sensitivity analysis) as confounders in the primary analyses? Smoking causes inflammation and elevated CRP, as does BMI, both of which have been associated with widespread differential DNA methylation in blood. So both of these meet the definition of confounders. And the authors' sensitivity analyses show a clear attenuation of effect sizes when BMI is included in the model (Fig. 3A) and when all covariates are included (Supplementary Fig. 6). To me, this indicates that these upstream factors are likely confounders of the CRP-DNA methylation associations, and adjusting for these in the main models would have been more appropriate. At a minimum, BMI and smoking are likely available for most of these cohorts. The authors
3. I don't understand what the goal of the correlation-clustering analysis is, or how it aids in the interpretation of the findings. The actual correlation structure is barely described... does 1 cluster represent CpGs that have correlated DNA methylation levels, and the other represent CpGs that aren't correlated? Or are they clustered by direction of correlation (positive vs. inverse)? Why is the RELA-binding site enrichment in cluster 2 interesting? None of the correlation cluster analyses is included in the discussion, and the interpretation is unclear. I suggest removing this from the manuscript.
4. The mediation analysis is also not well justified. The authors say that 6 different mediation models were considered, and 2 models were indicated for further exploration, then refers readers to the supplemental methods and tables. I can find no description of the 6 models in the supplemental materials, just a table (Supp Table 9) with poorly labeled columns that I cannot interpret. The information regarding mediation model selection should be very clearly stated within the main text of the manuscript, along with the assumptions that are being met or potentially violated – this information is critical for interpreting the mediation results. Also, the conceptual model is not clear. Given the results from the sensitivity analyses, isn't this model (CRP <- BMI -> DNA methylation; confounding) just as likely as this model (BMI -> CRP -> DNA methylation)? The author present results that provide supporting evidence for both confounding and mediation, but only focus on the mediation results in the interpretation/discussion.

Minor:

1. The results section would benefit from reporting of more actual statistics. The vast majority of results and statistics are sequestered into supplementals, which decreases the readability of this section. For each subsection, present the actual statistics for one or two of the most impactful results. Also, when first presenting the meta-analysis results, the authors should state which factors were controlled for in that analysis.

2. For each subsection in the results, this would be easier to understand if the goal of each analysis was clearly laid out before describing those results.
3. Overall, so many analyses were performed and presented, but they are mostly presented and interpreted separately. This paper would benefit from a much more integrated interpretation of the results, that identifies the intersections across different results that reinforce specific interpretations and inferences. This is done for the mediation and MR results, but not for the other findings.

Reviewer #2:

Remarks to the Author:

The manuscript by Wielscher reports the results of a trans-ethnic epigenome-wide association study (EWAS) on 22,774 individuals to identify CpGs associated with CRP levels, a proxy for low-grade inflammation. The 1,511 significant CpGs are enriched in transcription factor binding sites and genomic enhancer regions, suggesting that they may play a functional role. Mendelian randomization analysis suggest that CRP levels are causal for changes in DNA methylation and mediation analysis showed that obesity and smoking may drive these changes in DNA methylation. Epigenetic risk score was also calculated to enable estimation of adjusted relative risk of several disease traits based on the methylation signature.

Overall, this is an elegant and well executed study. It is highly significant because it was done in a multi-ethnic cohort and has relevance to a number of cardiorespiratory traits. A major strength of the study is the large sample size that allowed for identification of robustly associated loci across different ethnicities. The authors were able to use very stringent methods to identify the most robust associations and focused on CpGs that are independent from each other (they removed CpGs that had high correlations). All subsequent analyses were done very well and the results presented clearly.

I have two main issues with the manuscript:

1. While associations are robust statistically in terms of significance, there is no mention of effect sizes. Presumably, these are small effect sizes because they are in detected in blood? Effect sizes need to be discussed.
2. I am very concerned that the calculation of the adjusted relative risk score is misleading. The interpretation is that the full activation of the epigenetic risk score (all CpGs in the signature are fully methylated or unmethylated) increases the risk for MI by 20.3%, COPD by 5.6%, T2D risk by 11.3%, hypertension by 11.9 % and CAD by 29%. These are impressive numbers but we cannot base these calculations on fully methylated and fully unmethylated CpGs because that will never be the case in a mixed cell population. This is where the effect sizes observed need to be taken into account.

Response to Reviewers

We wish to thank the reviewers for their thoughtful comments, which we have considered carefully. We highlighted changed text sections in the main manuscript with yellow.

1. The authors applied genomic control to deal with population stratification and other unmeasured confounding. However, while GC does decrease inflation by restricting p-value thresholds, it does not correct for bias such as confounding. Since confounding is not addressed by GC, the authors should apply an alternative approach for addressing residual confounding that was specifically developed for addressing inflation and bias in EWAS (Controlling bias and inflation in epigenome- and transcriptome-wide association studies using the empirical null distribution | Genome Biology). This should be applied to cohort-specific results prior to fixed-effects meta-analysis.

We fully agree on the limitation of Genomic Control procedure in terms of GWAS as well as EWAS, especially when the outcome is associated to many small effects as expected in a typical EWAS setting. The reason for choosing Genomic Control approach in this study will be outlined in point two. The method suggested by Reviewer one uses a Bayesian approach to correct deviation of the test statistics from empirical null distribution as well as corrects for inflation by assuming a standard normal distribution rather than a chi square distribution. This makes the approach more flexible and well suited to correct inflation from regression analysis. Genomic Control does not account for test statistic bias and assumes a chi-square distribution when correcting for inflation. Following up on these facts we performed some analysis to detected possible bias in our test statistics and to determine if we have been too conservative when controlling for inflation. First, we inspected the distribution of each the test statistics contributing to our transethnic meta-analysis to evaluate their deviation from the empirical null distribution. Black line is the expected distribution of Z-scores, red line is the observed distribution of Z-scores.

Revision Figure 1: Overview of distributions of Z-scores across studies on 450K or EPIC platform in our transeethnic meta-analysis. Black line represents empirical null distribution. Red line represents observed distribution.

Additionally, we measured lambda and bias (i.e. genomic inflation) of the raw data alongside with corrected genomic inflation values by the method suggested (Reviewer 1). Those values are given in Revision table 1.

STUDY	BIAS	LAMBDA	LAMBDA CORRECTED
AIRWAVE	-0.033	1.05	1.02
ARIC	0.334	2.27	1.31
ARIES	0.208	1.18	1.03
BHS.B	0.238	0.88	0.91
BHS.W	0.008	0.94	0.95
BIOS.CODAM	-0.166	1.06	1.00

BIOS.LLS	0.003	1.04	1.02
BIOS.NTR	-0.089	1.33	1.09
BIOS.PAN	0.185	1.09	1.00
CARDIOGENICS	0.011	1.04	1.01
CHS.B	-0.030	1.17	1.05
CHS.W	0.188	1.15	1.01
EGCUT.CTG	-0.011	1.04	1.01
EPIC.NORFOLK	0.003	1.22	1.10
EPICOR	-0.247	1.12	1.01
ESTHER.1A	-0.036	1.62	1.22
ESTHER.1B	-0.026	1.19	1.08
FHS	-0.016	1.11	1.04
GENOA	0.373	1.55	1.14
KORA	-0.001	1.16	1.07
LBC	0.040	0.86	0.92
LLD	-0.004	1.13	1.05
LOLIPOP	0.008	1.25	1.10
NAS	0.046	0.94	0.91
NFBC1966	-0.017	1.25	1.10
NFBC1986	-0.003	1.11	1.04
ROTTERDAM	0.019	1.62	1.16
SHIP	0.004	1.12	1.04
YFS	-0.003	1.02	1.01

Revision Table 1: Overview of Bias, Inflation and corrected Inflation (using method suggested by Reviewer 1) across all studies on 450K or EPIC platform.

This initial analysis (Revision Figure 1, Revision Table 1) showed that the test statistics of the majority of studies did not deviate from empirical null distribution (Revision Figure 1, red line vs black line). Only in very few studies such as ARIC, BHS.B or GENOA correcting the test statistic bias will have an effect on their contribution to our transethnic meta-analysis. Thus, as next step we performed a meta-analysis of the studies on 450K and EPIC arrays without genomic control but with corrected test statistic values from the studies. In our hands the R-package (“BACON”) associated to the publication “Controlling bias and inflation in epigenome- and transcriptome-wide association studies using the empirical null distribution” did only return meta-analyzed values if values were present in all participating studies. This reduced the number of analyzed CpGs to 319640 out of 405019. A total 211 CpG markers reported as significantly associated to CRP (n=1765 CpG markers) in the presented study had at least one NA (not available) value in one of the 22 participating studies. We find 1471 CpGs out of 1554 CpGs were present in both meta-analysis and genome-wide significant in both studies. 83 CpGs had an adjusted P-value smaller than 2×10^{-6} . Within the set of 1554 CpG present the highest adjusted P-value observed was 1.9×10^{-6} . For the association to serum CRP our initial strategy does not seem overly stringent: In total our transethnic meta-analysis yielded 1765 markers out of 405019 analyzed sites. For comparison, the new meta-analysis would have yielded around 2046 markers. This suggest sensitivity and specificity of these two meta-analysis methods are similar for this study/exposure.

Revision Figure 2: Upset plot shows the overlaps between meta-analysis using genomic control and meta-analysis using test-statistic bias correction method. CRP EWAS is the complete lists of all markers presented in the study (n=1765). This includes correlated markers, which were removed from downstream analysis in the manuscript. “bias_corrected_sig” is the list of markers with a P value smaller than 1xE-07 from bias corrected method. “bias_corrected_E06” are all markers, significant in genomic control meta-analysis and showing a P value of smaller than 1xE-06 in bias corrected meta-analysis. “bias_corrected_2xE06” same as above with threshold relaxed to 2xE-06. “bias_corrected_NA” are marker not available in current bias corrected meta-analysis.

In summary, we find that correction for test statistic bias and calculation of lambda inflation using the actual distribution of the test statistics to calculate lambda values are very important steps forward in EWAS analysis. However, they seem to play a minor role in this study/this exposure.

Nevertheless, we added a table with all genome wide markers from bias corrected meta-analysis to our supplemental tables database and included an additional column giving the adjusted P-values from this new meta-analysis to supplemental table 2. Furthermore, the method including plots that show distribution of Z-scores and the Upset plot comparing the two approaches are discussed in our results section of new version of the manuscript.

Page 8:

“Additionally, we applied an alternative strategy to control for bias and inflation (Supplemental Figure 8, 9 Supplemental table 2). Using this approach P-values were ranging from 2.2E-124 to 1.9xE-06, and we discovered a total of 144 additional CRP associated markers given in Supplemental table 3.”

2. However, the best approach is to control for confounding within the regression model itself, as long as the appropriate data are available, which raises another question. Why did the authors not adjust for BMI and smoking (or other variables in the sensitivity analysis) as confounders in the primary analyses? Smoking causes inflammation and elevated CRP, as does BMI, both of which have been associated with widespread differential DNA methylation in blood. So both of these meet the definition of confounders. And the authors' sensitivity analyses show a clear attenuation of effect sizes when BMI is included in the model (Fig. 3A) and when all covariates are included (Supplementary Fig. 6). To me, this indicates that these upstream factors are likely confounders of the CRP-DNA_m associations, and

adjusting for these in the main models would have been more appropriate. At a minimum, BMI and smoking are likely available for most of these cohorts.

Blood CRP levels are influenced by a large variety of measured and unmeasured factors that in turn enhance and alter many clinically relevant parameters in the human body. Thus, rather than focusing straightway on factors we also thought are important, we present associations between CRP levels and blood DNA methylation as they are. Of course, as pointed out by Reviewer 1 that increases the risk of reporting false positive associations, this is why we choose a rather conservative approach to control for inflation. Due to these complex relationships our analysis and writing group had intensive discussion about the analytical strategy in the planning phase of the analysis. However, the approach and rationale behind our analysis strategy was obviously not clearly enough indicated in the manuscript. We have now added a section where we explain the rationale of our approach. (Discussion, p 19-20) We hope this makes our motivation for the mediation analysis clearer as well.

“Our analysis strategy can be summarized as follows: To maximize power in this analysis we use a sparse regression model including only technical confounders. Next, we apply genomic control procedure, a conservative approach more typically performed in genome wide association studies¹ (Figure 1B), to prevent from false positive associations between blood CRP levels and DNA methylation. This allows us to use the maximum available sample size in the regression analyses that can be taken forward in the downstream analyses. Within the study we then work towards a better understanding of the driving forces of the CRP DNA methylation association. We present Mediation and Mendelian Randomization analysis to understand the potential causal connection between DNA methylation, blood CRP levels, BMI and smoking. Finally, we present a DNA methylation risk score analysis, without any underlying assumptions, based on all CRP associated CpGs alongside with more fine-grained analysis stratified by the analysis results of the study (Supplemental table 2).”

For your reference we summarized the sensitivity analysis in Revision table 2. Also, we calculated correlation coefficients of Z scores between the unadjusted base model and each model of the sensitivity analysis.

Model	min. N in model	max. N in model	changed direction of effect	ρ(Z-scores)	overlap to published studies
BMI	19357	22639	0	0.985	120
Smoking	3892	4989	9	0.988	690
Waist	3900	4998	6	0.975	NA
Hip	3901	4999	1	0.983	NA

total cholesterol	3920	5018	4	0.988	NA
triglycerides	3917	4751	5	0.985	NA
insulin	3916	4998	2	0.983	NA
all risk factors	3762	4576	18	0.964	NA

Revision Table 2: Sensitivity analysis was performed on 4 cohorts: AIRWAVE, LOLIPOP, NFBC1966 and NFBC1986. Covariates were added one at the time to the base model. We calculated a person rho to reflect correlation between Z-scores of the base model and each model in Revision table 2.

In total we found 22 markers that showed a changed direction of effect in one or more of the sensitivity analyses. Out of these 22 markers 18 flagged as an independent locus and thus were further analyzed in the presented study. We hope that the general accordance of Z-values, high correlation values between Z-scores and a very small fraction of CpGs (18 out of 1511) showing a changed direction of effect is sufficiently justifying our decision to start our analysis on a minimalistic regression model.

3. I don't understand what the goal of the correlation-clustering analysis is, or how it aids in the interpretation of the findings. The actual correlation structure is barely described... does 1 cluster represent CpGs that have correlated DNA methylation levels, and the other represent CpGs that aren't correlated? Or are they clustered by direction of correlation (positive vs. inverse)? Why is the RELA-binding site enrichment in cluster 2 interesting? None of the correlation cluster analyses is included in the discussion, and the interpretation is unclear. I suggest removing this from the manuscript.

The calculation of correlation values is based on actual DNA methylation beta values recorded in 4 studies: AIRWAVE, KORA, NFBC1966 and NFBC1986. Thus, the presented correlation clusters include CpGs upregulated and downregulated by blood CRP levels. The clustering then is based on correlation values, which however does not mean that all positive vs. inverse correlated CpGs represent one cluster (Figure 2A). Both clusters consist of positive and negative correlated CpGs, however the pattern of across all analyzed CpGs is the same for each cluster.

We had the idea for this analysis when looking at the overall correlation pattern of CRP associated CpG. We noticed a clear pattern with these correlations (Figure 2A). More strikingly this pattern was conserved within the four analyzed cohorts (Supplemental Figure 5). These four cohorts were spanning a median age from 16 to 61 years and the data were produced on EPIC and 450K chips, thus we were confident that this is not a technical artefact. Following up on this, the best analysis we could come up with to further investigate these patterns was to group them and stratify our list of CRP associated loci according to these patterns. We included this in our enrichment analysis as well as our risk score analysis (supplemental table 22) and further described the clusters in line 345 to 357.

Briefly, the clusters seem to have a very similar effect on clinically relevant outcomes. GO term (Gene ontology terms) enrichment showed enrichment in immune system processes and metabolic processes, which were enriched to larger or different extend in cluster 2 compared to cluster one. We observed the most pronounced differences between correlation clusters

within RELA binding site methylation (NF- κ Bp65 subunit, Figure 4). GO term enrichment showed enrichment of Immune system processes for both clusters and enrichment of metabolic processes however only for cluster 2 (Figure 4). A recent study suggests mutual regulation of overnutrition and inflammation by NF- κ B², which fits very well to the GO term enrichments observed for cluster 2. Similarly, other studies showed that RELA binding sites are not only controlling Immune system associated genes but can also control triglyceride levels and lipogenesis^{3,4}. Finally, the vast majority of RELA binding sites is linked to epigenetic regulators² such H3K27ac, H3K4me both of which are closely linked to changes in DNA methylation^{5,6}. Thus, we speculate that cluster 2 might reflect this mutual regulation of overnutrition and inflammation.

We added the following to the Discussion section of the text:

“Our cluster analysis of correlation coefficients showed two distinct groups. We observed the most pronounced differences between correlation clusters within RELA binding site methylation (NF- κ Bp65 subunit, Figure 4). GO term enrichment showed enrichment of Immune system processes for both clusters and enrichment of metabolic processes for cluster 2 (Figure 4). A recent study suggests mutual regulation of overnutrition and inflammation by NF- κ B², which fits very well to the GO term enrichments observed for cluster 2. Similarly, other studies showed that RELA binding sites are not only controlling Immune system associated genes but can also control triglyceride levels and lipogenesis^{3,4}. Finally, the vast majority of RELA binding sites is linked to epigenetic regulators² such H3K27ac, H3K4me both of which are closely linked to changes in DNA methylation^{5,6}. Thus, we speculate that cluster 2 might reflect this mutual regulation of overnutrition and inflammation.”

Even if this remains speculative, without any in vitro experiments, we think it is important to draw attention to those patterns in the data. It should encourage other researchers to investigate this and serves as reference, as we observed this pattern across almost 5000 participants. Furthermore, our understanding of the genome including its 3D structures steadily increases thus we might find an even simpler explanation for this pattern in the near future.

4. The mediation analysis is also not well justified. The authors say that 6 different mediation models were considered, and 2 models were indicated for further exploration, then refers readers to the supplemental methods and tables. I can find no description of the 6 models in the supplemental materials, just a table (Supp Table 9) with poorly labeled columns that I cannot interpret. The information regarding mediation model selection should be very clearly stated within the main text of the manuscript, along with the assumptions that are being met or potentially violated - this information is critical for interpreting the mediation results. Also, the conceptual model is not clear. Given the results from the sensitivity analyses, isn't this model (CRP \leftarrow BMI \rightarrow DNA methylation; confounding) just as likely as this model (BMI \rightarrow CRP \rightarrow DNA methylation)? The author present results that provide supporting evidence for both confounding and mediation, but only focus on the mediation results in the interpretation/discussion.

We apologize for the description of the 6 initial Mediation models being lost in the process. We corrected the reference to supplemental table 9 and the mislabel in line 190 in the method section. We also updated supplemental table 9 and labeled the columns accordingly. For your reference we summarize the analyzed models here:

- ❖ Model 1: BMI -> CRP -> DNAmethylation
- ❖ Model 2: CRP -> BMI -> DNAmethylation
- ❖ Model 3: SMOKING -> CRP -> DNAmethylation
- ❖ Model 4: BMI -> DNAmeth -> CRP
- ❖ Model 5: CRP -> DNAmeth -> BMI
- ❖ Model 6: SMOKING -> DNAmeth -> CRP

	path C	path C'	path A	path B	negative indirect effect	positive indirect effect	significant mediation	positive effect
model 1	436	181	1511	562	334	102	177	10
model 2	734	715	0	583	561	173	0	0
model 3	724	662	1511	181	507	217	65	9
model 4	1511	1511	436	562	364	1389	0	64
model 5	1511	1511	724	181	474	1279	0	22
model 6	0	0	734	583	NA	NA	NA	NA

Revision Table 3: Overview of model selection process in Mediation analysis.

In our Mediation analysis as well as while evaluating which initial Mediation Analysis models to take forward we followed Baron and Kenny's steps for mediation:

Coefficient of path C is statistically significant that means the independent variable (for model 1: BMI) is a significant predictor of the dependent variable (for model 1: DNAmethylation).

Coefficient of path A is statistically significant, that means the mediator (for model 1: CRP) is significantly associated to the independent variable (for model 1: BMI). If that is not the case the mediator cannot mediate anything.

Coefficient of path B is statistically significant that means the mediator (for model 1: CRP) is associated to the dependent variable (for model 1: DNA methylation) while adjusting for the independent variable (for model 1: BMI).

The initial 6 models were chosen to inform about directionality. In other words to make sure we did identify the mediator as mediator and not, for example, mistake an independent variable as mediator.

For evaluation of the 6 models, we evaluated if one or more assumption for Mediation analysis according to Baron Kenney were violated. As given in on Revision table 3 (Supplemental table 9) we did not find any significant association in path A for model 2; model 4 and 5 did not have a negative indirect effects and model 6 did not produce results, because path C association between Smoking and CRP was not significant in NFBC1966. That left only model 1 and model 3 that did not violate assumption necessary to perform a meaningful mediation analysis.

Additionally, incorporating published literature into the model selection process we were looking into studies that showed that BMI is causal for increased levels of blood CRP^{7,8}, on a global level BMI causes alterations in DNA methylation patterns⁹ as well as CRP causing changes in alteration in DNA methylation pattern (this study). This makes model 2, 4, 5 and 6 highly unlikely as at least one of their path according to Baron Kenny is in conflict with the findings of these studies.

We updated the concordant section in the results (section Driving forces of CRP CpG association: Mediation Analysis p15):

“To avoid violation of assumption made by Baron Kenny approach we restricted the analysis to loci with a nominal significant C-path and A-path (Figure 3) and only report mediation if the indirect effect (C'-path – C-path) is negative.”

“Model 1: BMI -> CRP -> DNAmethylation;

Model 2: CRP -> BMI -> DNAmethylation;

Model 3: SMOKING -> CRP -> DNAmethylation;

Model 4: BMI -> DNAmeth -> CRP; Model 5: CRP -> DNAmeth -> BMI;

Model 6: SMOKING -> DNAmeth -> CRP (supplemental results table 1).

Also, we added the model selection process to the method section (p6-p7).

For model selection, we evaluated if one or more assumption for Mediation analysis were violated. As given in Supplemental table 10 we did not find any significant association in path A for model 2; model 4 and 5 did not have a negative indirect effects and model 6 did not produce results, because path C association between Smoking and CRP was not significant in NFBC1966. That left only model 1 and model 3 that did not violate assumption necessary to perform a meaningful mediation analysis. Additionally, incorporating published literature into the model selection process we were looking into studies that showed that BMI is causal for increased levels of blood CRP^{7,8}, on a global level BMI causes alterations in DNA

methylation patterns⁹ as well as CRP causing changes in alteration in DNA methylation pattern (this study). This makes model 2, 4, 5 and 6 highly unlikely as at least one of their path according to Baron Kenny is in conflict with the findings of these studies.”

For the presented CRP associated DNA methylation signature we cannot completely rule out confounding in a sense that BMI influences both serum CRP levels and DNA methylation level. We can however say that confounding seems highly unlikely for the 729 CpGs where we show a Mediation of BMI via CRP to DNA methylation (model 1, nominal significant). In case of confounding by BMI path B (DNAmeth ~ CRP + BMI) association would have been abrogated by adding BMI as covariate. For the remaining CpGs presented we can only evaluate the findings of the sensitivity analysis (Revision table 2)

Minor:

1. The results section would benefit from reporting of more actual statistics. The vast majority of results and statistics are sequestered into supplementals, which decreases the readability of this section. For each subsection, present the actual statistics for one or two of the most impactful results. Also, when first presenting the meta-analysis results, the authors should state which factors were controlled for in that analysis.

In section Transethnic discovery we added:

“Our regression model was adjusted for age, sex and blood cell type composition and technical confounders.”

In section Transethnic discovery we added:

“Effect sizes (supplemental Figure 211) were ranging from 0.50 ± 0.06 to 9.85 ± 0.29 logarithmic mg/L change in CRP per unit increase in DNA methylation in blood (scale for methylation 0-1, where 1 represents 100% methylation) P values were ranging from 9.9×10^{-8} to 1.9×10^{-69} .”

In section Sensitivity analysis we added:

“For BMI adjusted model effect sizes were ranging from 0.241 ± 0.151 to 6.648 ± 0.482 logarithmic mg/L change in CRP per unit increase in DNA methylation in blood with consistent direction of effect.”

2. For each subsection in the results, this would be easier to understand if the goal of each analysis was clearly laid out before describing those results.

According your suggestion we updated several paragraphs:

In section Sensitivity analysis we added:

“To evaluate the stability of the presented association results, we compared the base model to a model additionally adjusted for BMI across all studies”

In section Driving forces of CRP CpG association: Mediation Analysis we added:

“Sensitivity showed that BMI attenuates but not abrogates the effect of CRP on DNA methylation (Figure 3) and a high overlap to published BMI associated CpGs. We observed a similar situation for smoking (Supplemental table 4). To better understand those relationships between these traits and our data, we performed a mediation analysis”

3. Overall, so many analyses were performed and presented, but they are mostly presented and interpreted separately. This paper would benefit from a much more integrated interpretation of the results, that identifies the intersections across different results that reinforce specific interpretations and inferences. This is done for the mediation and MR results, but not for the other findings.

We added a paragraph about the enrichment and correlation analysis to the Discussion section:

“Our cluster analysis of correlation coefficients showed two distinct groups. We observed the most pronounced differences between correlation clusters within RELA binding site methylation (NF- κ Bp65 subunit, Figure 4). GO term enrichment showed enrichment of Immune system processes for both clusters and enrichment of metabolic processes for cluster 2 (Figure 4). A recent study suggests mutual regulation of overnutrition and inflammation by NF- κ B², which fits very well to the GO term enrichments observed for cluster 2. Similarly, other studies showed that RELA binding sites are not only controlling Immune system associated genes but can also control triglyceride levels and lipogenesis^{3,4}. Finally, the vast majority of RELA binding sites is linked to epigenetic regulators² such H3K27ac, H3K4me both of which are closely linked to changes in DNA methylation^{5,6}. Thus, we speculate that cluster 2 might reflect the mutual regulation of overnutrition and inflammation.”

Reviewer #2 (Remarks to the Author):

1. While associations are robust statistically in terms of significance, there is no mention of effect sizes. Presumably, these are small effect sizes because they are not detected in blood? Effect sizes need to be discussed.

Due to large sample size and the fact that we measure our signal in blood subsequent to adjusting for cell type composition in blood effect sizes are quite small. It is challenging to compare those subtle changes in DNA methylation to landslide like changes as observed in carcinogenesis or when comparing different tissues or disease states. To draw attention to effect sizes observed in our study we made a volcano plot (Supplemental Figure 2) and also included ranges and examples of actual effect size estimates in our updated manuscript. Finally, in the new version of the manuscript we present standardized regression coefficients. Briefly, we had the standard errors and mean values of every CpG recorded across all participating cohorts. Using the mean values and samples numbers we could combine the Standard errors across all cohorts. Following the same strategy and using lnCRP data from 6 cohorts (n=7403), we combined standard errors for lnCRP values. This allowed us to calculate standardized regression coefficients $\text{Coeff}_{\text{standardized}} = (\text{Coeff}_{\text{CRP} \sim \text{CpG} + [\dots]} * \text{combined } \text{SD}_{\text{CpG}}) / \text{combined } \text{SD}_{\text{lnCRP}}$. Standardized regression coefficient for every marker presented in the study are given in Supplemental table 2, calculation is explained in method section page 2 and we now give examples of standardized effect sizes in the main text of the manuscript.

Volcano plot:

Supplemental Figure 2: Volcano plot CpG methylation and serum CRP association result. Red dots were taken forward to further analysis in the current study. Dotted horizontal line is P-value threshold 1xE-07. Vertical line indicates smallest observed effect size of CpG in analysis. Effect size is logarithmic ml/L change in CRP per unit increase in DNA methylation.

Range of observed effect sizes and standardized regression coefficients

In section Transethnic discovery we added:

“Effect sizes range from 0.50 ± 0.06 to 9.85 ± 0.29 logarithmic mg/L change in CRP per unit increase in DNA methylation in blood (scale for methylation 0-1, where 1 represents 100%

methylation). We also standardized the regression coefficients from the trans-ethnic discovery, for example, one CpG near NF-κB inhibitor epsilon (NFKBIE) showed a decrease of 8 standard deviations of DNA methylation per standard deviations of logarithmic CRP. Standardized regression coefficients were ranging from 22.2 for a CpG near STK40 to -21.5 for a CpG not in vicinity of any gene (Supplemental table 2).”

In Section Sensitivity analysis we added:

“For BMI adjusted model effect sizes were ranging from 0.241±0.151 to 6.648±0.482 logarithmic mg/L change in CRP per unit increase in DNA methylation in blood with consistent direction of effect”

A table giving the number of CpGs with changed direction of effect observed in the Sensitivity analysis

MODEL	MIN. N IN MODEL	MAX. N IN MODEL	CHANGED DIRECTION OF EFFECT	SIGNIFICANT DIFFERENCE IN COEFFICIENT*	RHO OF Z SCORES	OVERLAP TO PUBLISHED STUDIES
BMI	19357	22639	0	29	0.985	120
SMOKING	3892	4989	9	40	0.988	690
WAIST	3900	4998	6	27	0.975	NA
HIP	3901	4999	1	20	0.983	NA
TOTAL CHOLESTEROL	3920	5018	4	49	0.988	NA
TRIGLYCERIDES	3917	4751	5	25	0.985	NA
INSULIN	3916	4998	2	18	0.983	NA
ALL RISK FACTORS	3762	4576	18	91	0.964	NA

We updated supplemental table 5. It gives the number of markers that changed the direction of effect in the sensitivity analysis. We hope that these modifications give the reader an impression of the effect sizes observed for the reported CRP DNA methylation association in blood.

2. I am very concerned that the calculation of the adjusted relative risk score is misleading. The interpretation is that the full activation of the epigenetic risk score (all CpGs in the signature are fully methylated or unmethylated) increases the risk for MI by 20.3%, COPD by 5.6%, T2D risk by 11.3%, hypertension by 11.9 % and CAD by 29%. These are impressive numbers but we cannot base these calculations on fully methylated and fully unmethylated CpGs because that will never be the case in a mixed cell population. This is where the effect sizes observed need to be taken into account.

To avoid presenting these finding out of context we removed this section from the abstract.

We also added some additional explanation to the concordant section of the result section.

“This may be interpreted as follows: A full activation of the CRP DNA methylation risk score increases the risk for myocardial infarction by 20.3% (Figure 5A), indicating that if all discovered CpG loci would be either fully methylated or unmethylated according their direction of CRP association to have this impact on myocardial infarction. A full activation of the discovered CpG signature is, however, very unlikely to happen for one single individual, thus these estimates represent the maximum theoretical impact of the discovered CpG on these clinically relevant traits.”

References

1. Yang, J. *et al.* Genomic inflation factors under polygenic inheritance. *Eur J Hum Genet* **19**, 807-12 (2011).
2. Kracht, M., Muller-Ladner, U. & Schmitz, M.L. Mutual regulation of metabolic processes and proinflammatory NF-kappaB signaling. *J Allergy Clin Immunol* **146**, 694-705 (2020).
3. Gruber, P.J., Torres-Rosado, A., Wolak, M.L. & Leff, T. Apo CIII gene transcription is regulated by a cytokine inducible NF-kappa B element. *Nucleic Acids Res* **22**, 2417-22 (1994).
4. Baker, R.G., Hayden, M.S. & Ghosh, S. NF-kappaB, inflammation, and metabolic disease. *Cell Metab* **13**, 11-22 (2011).
5. Charlet, J. *et al.* Bivalent Regions of Cytosine Methylation and H3K27 Acetylation Suggest an Active Role for DNA Methylation at Enhancers. *Mol Cell* **62**, 422-431 (2016).
6. Okitsu, C.Y. & Hsieh, C.L. DNA methylation dictates histone H3K4 methylation. *Mol Cell Biol* **27**, 2746-57 (2007).
7. Timpson, N.J. *et al.* C-reactive protein levels and body mass index: elucidating direction of causation through reciprocal Mendelian randomization. *Int J Obes (Lond)* **35**, 300-8 (2011).
8. Welsh, P. *et al.* Unraveling the directional link between adiposity and inflammation: a bidirectional Mendelian randomization approach. *J Clin Endocrinol Metab* **95**, 93-9 (2010).
9. Wahl, S. *et al.* Epigenome-wide association study of body mass index, and the adverse outcomes of adiposity. *Nature* **541**, 81-86 (2017).

Reviewers' Comments:

Reviewer #1:

Remarks to the Author:

I commend the authors on a thorough and satisfactory rebuttal. The authors have addressed all of my comments.

However, I share the same concern as reviewer #2: "The adjusted relative risk score is misleading. The interpretation is that the full activation of the epigenetic risk score (all CpGs in the signature are fully methylated or unmethylated) increases the risk for MI by 20.3%, COPD by 5.6%, T2D risk by 11.3%, hypertension by 11.9 % and CAD by 29%. These are impressive numbers but we cannot base these calculations on fully methylated and fully unmethylated CpGs because that will never be the case in a mixed cell population. This is where the effect sizes observed need to be taken into account".

While the authors have added a sentence of caution in interpreting these results, I do not think that is sufficient. It is likely impossible that all CpGs in this risk score could transition from completely methylated to completely unmethylated, or vice versa. So the value of these reported statistics are unclear. The authors should re-visit this analysis and base the score on DNA methylation differences that are at least plausible. Consider using a 1 IQR increase/decrease in methylation, based on the distribution of beta-values for each CpG, from one of the largest cohorts involved in the study.

Aside from the above concern, I think this is an excellent revised manuscript and the results will be of interest to the broad readership of Nat Comms.

Reviewer #2:

Remarks to the Author:

The authors did a very thorough job of responding to reviewer comments, including running a number of additional analyses. I think this made the manuscript stronger and I have no additional suggestions for improvement.

Response to Reviewers

From reviewer: I commend the authors on a thorough and satisfactory rebuttal. The authors have addressed all of my comments.

However, I share the same concern as reviewer #2: "The adjusted relative risk score is misleading. The interpretation is that the full activation of the epigenetic risk score (all CpGs in the signature are fully methylated or unmethylated) increases the risk for MI by 20.3%, COPD by 5.6%, T2D risk by 11.3%, hypertension by 11.9 % and CAD by 29%. These are impressive numbers but we cannot base these calculations on fully methylated and fully unmethylated CpGs because that will never be the case in a mixed cell population. This is where the effect sizes observed need to be taken into account".

While the authors have added a sentence of caution in interpreting these results, I do not think that is sufficient. It is likely impossible that all CpGs in this risk score could transition from completely methylated to completely unmethylated, or vice versa. So the value of these reported statistics are unclear. The authors should re-visit this analysis and base the score on DNA methylation differences that are at least plausible. Consider using a 1 IQR increase/decrease in methylation, based on the distribution of beta-values for each CpG, from one of the largest cohorts involved in the study.

Thank you for the constructive criticism. Based on the combined means and standard error or all cohorts we estimated the quartiles of DNA methylation beta values of the CpGs involved in the score. We plotted a distribution of the involved IQR for your reference.

Figure 1.: Distribution of inter quartile ranges (IQR) of CpG involved in DNA methylation risk score calculation. The IQR were estimated from combined mean and standard error across all cohorts that participated in this study.

We understand it would ease interpretation to give the estimates per IQR of actual DNA methylation values of selected CpGs or a mean IQR of CpGs involved in the risk score calculation. This is, however, not possible as we scaled the DNA methylation scores to make them better comparable between studies and traits. What we did to follow up on your advice is

1. we had a closer look at the distribution of score value in one cohort (Figure 1).
2. We adjusted the estimates for the whole range of scores and now give an estimate per 1% changed methylation in the DNA methylation risk score.

Figure 2: Distribution of DNA methylation risk score in NFBC1966. Red is interquartile range. For adjustment of the relative risk we used the whole range of the score to produce estimates per percent changed DNA methylation risk score

The new adjusted relative risks were calculated after adjusting the logOdds ratio from logistic regression in the score analysis by the range of observed scores. We found 1.007% increase of the relative risk for COPD by percent change in DNAmeth risk score. For T2D we observed increase of 1.7%, for MI 2.9%, CAD 4.3% and for hypertension 0.2 % by percent increase in DNA methylation risk score.

We updated the text in the result section:

“This may be interpreted as follows: A full activation of the CRP DNA methylation risk score as given in Figure 5A indicates theoretical maximum impact of the discovered CpG on these clinically relevant traits. For this the CRP associated CpGs would be either have to be fully methylated or unmethylated according their direction of effect. This is, however, very unlikely to happen for one single individual. Thus, we also calculated the risk conveyed by one percent change in the DNA methylation risk score. The relative risk increase per one percent increased DNA methylation risk score was 1.007% for COPD, 1.7% for T2D, 2.9% for myocardial infarction 4.3% coronary artery disease and 0.2% for hypertension.”

We also updated the Legend of Figure 5

“Those estimates indicate the theoretical maximum impact of the discovered CpG signature (100% DNA methylation change) on the tested traits. The risk conveyed by one percent change in the DNA methylation risk score on the tested traits was 1.007% for COPD, 1.7% for T2D, 2.9% for myocardial infarction 4.3% coronary artery disease and 0.2% for hypertension.”

Reviewers' Comments:

Reviewer #1:

Remarks to the Author:

I thank the authors for making this additional revision. They addressed this concern, and previous concerns beautifully. This is an excellent manuscript.